# A Quadratic Synchronization Rule for Distributed Deep Learning

**Xinran Gu**[1*]   **Kaifeng Lyu**[4*]   **Sanjeev Arora**[4†]   **Jingzhao Zhang**[1,2,3†]   **Longbo Huang**[1†]

[1]Institute for Interdisciplinary Information Sciences, Tsinghua University
[2]Shanghai Qizhi Institute        [3]Shanghai AI Laboratory
[4]Department of Computer Science & Princeton Language and Intelligence, Princeton University
`gxr21@mails.tsinghua.edu.cn    {klyu,arora}@cs.princeton.edu`
`{jingzhaoz,longbohuang}@tsinghua.edu.cn`

## Abstract

In distributed deep learning with data parallelism, synchronizing gradients at each training step can cause a huge communication overhead, especially when many nodes work together to train large models. Local gradient methods, such as Local SGD, address this issue by allowing workers to compute locally for $H$ steps without synchronizing with others, hence reducing communication frequency. While $H$ has been viewed as a hyperparameter to trade optimization efficiency for communication cost, recent research indicates that setting a proper $H$ value can lead to generalization improvement. Yet, selecting a proper $H$ is elusive. This work proposes a theory-grounded method for determining $H$, named the Quadratic Synchronization Rule (QSR), which recommends dynamically setting $H$ in proportion to $\frac{1}{\eta^2}$ as the learning rate $\eta$ decays over time. Extensive ImageNet experiments on ResNet and ViT show that local gradient methods with QSR consistently improve the test accuracy over other synchronization strategies. Compared with the standard data parallel training, QSR enables Local AdamW on ViT-B to cut the training time on 16 or 64 GPUs down from 26.7 to 20.2 hours or from 8.6 to 5.5 hours and, at the same time, achieves $1.12\%$ or $0.84\%$ higher top-1 validation accuracy.

## 1 Introduction

The growing scale of deep learning necessitates distributed training to reduce the wall-clock time. Data parallel training is a foundational technique that distributes the workload of gradient computation to $K$ workers, also serving as a key building block of more advanced parallel strategies. At each step of this method, each worker first computes gradients on their own local batches of data. Then, they take an average over local gradients, which typically involves a costly All-Reduce operation. Finally, they update the model parameter with the averaged gradient and a gradient-based optimizer OPT, e.g., SGD, AdamW. In this paper, we term the data parallel implementation of optimizer OPT as "Parallel OPT". See Algorithm 1 for the pseudocode. The cost for this data parallelism is obvious. Frequent gradient synchronization can induce huge communication overhead as the number of workers and model size grow, severely hindering the scalability of distributed training (Tang et al., 2021; Li et al., 2022; Xu et al., 2023).

One approach to reducing this communication overhead is Local SGD (Stich, 2018; Zhou & Cong, 2018; Woodworth et al., 2020). Rather than synchronizing gradients at every step, Local SGD allows workers to independently train their local replicas using their own local batches with SGD updates. It is only after completing $H > 1$ local steps that these workers synchronize, where the model parameters get averaged over all replicas. Notably, while we mention SGD, this approach can be readily adapted to other popular optimizers. In this paper, if a gradient-based optimizer OPT is used for local updates, we term the variant as "Local OPT" (e.g., Local SGD, Local AdamW), and collectively refer to this class of approaches as *local gradient methods*. We provide a pseudocode for local gradient methods in Algorithm 2.

The main focus of this paper is to study the best strategies to set the synchronization period $H$ (i.e., the number of local steps per communication round) in local gradient methods. While setting $H$ to a larger value reduces communication, a very large $H$ can hinder the training loss from decreasing

---

*Equal contribution
†Corresponding authors

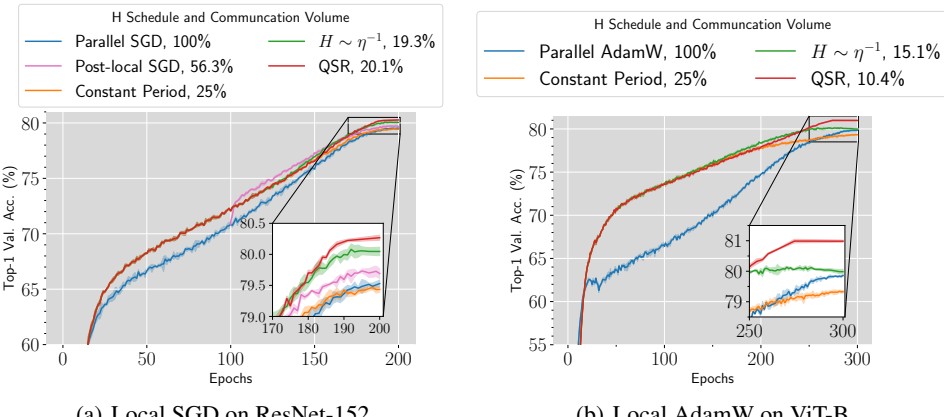

(a) Local SGD on ResNet-152          (b) Local AdamW on ViT-B

Figure 1: When training ResNet-152 and ViT-B on ImageNet with cosine learning rate decay, Local SGD/AdamW with QSR consistently outperforms data parallel methods or Local SGD/AdamW with other synchronization strategies in terms of top-1 validation accuracy, while only requiring 20.1% and 10.4% of the communication volume, respectively. With QSR, Local SGD on ResNet or Local AdamW on ViT cuts the training time from 20.7 to 18 hours or 26.7 to 20.2 hours on 16 GPUs, when compared with data parallel methods. We report the mean and the standard deviation over 3 runs. See Appendix C for training details.

at normal speed, since the local replicas may significantly diverge from each other before averaging. Indeed, it has been observed empirically that larger $H$ leads to higher training loss after the same number of steps (Wang & Joshi, 2021; Ortiz et al., 2021), and efforts to analyze the convergence of local gradient methods in theory usually end up with loss bounds increasing with $H$ (Khaled et al., 2020; Stich, 2018; Haddadpour et al., 2019; Yu et al., 2019). To better trade-off between communication cost and optimization speed, Kamp et al. (2014); Wang & Joshi (2019); Haddadpour et al. (2019); Shen et al. (2021) proposed adaptive synchronization schemes, such as linearly increasing $H$ as the iteration goes on (Haddadpour et al., 2019), or adjusting $H$ based on the variance in model parameters (Kamp et al., 2014). Nonetheless, their effectiveness has only been validated on linear models or small-scale datasets, e.g., CIFAR-10/100.

All these strategies are developed to avoid sacrificing too much training loss, but training loss is *never* the final evaluation metric that one cares about in deep learning. Due to the overparameterized nature of modern neural networks, reaching the same training loss does not correspond to the same performance on test data. It has also been long known that the choice of optimizers or hyperparameters can change not only the optimization speed of the training loss but also their *implicit bias* towards solutions with different test accuracies.

The presence of this implicit bias indeed complicates the picture of setting $H$ in local gradient methods. Though a large $H$ might be harmful for training loss, it has been observed empirically that setting $H$ properly can sometimes improve rather than hurt the final test accuracy. Lin et al. (2020) are the first to report this phenomenon. Comparing with running just the standard data parallel SGD (equivalent to $H = 1$), they observed that switching from SGD to Local SGD ($H > 1$) halfway through consistently leads to higher final test accuracy. Local SGD with this specific schedule of $H$ is designated as *Post-local SGD*. Lin et al. (2020)'s work opens up a new angle in setting $H$ in local gradient methods, yet, the proposed schedule in Post-local SGD, referred to as the post-local schedule in this paper, is suboptimal in improving test accuracy. It was later reported by Ortiz et al. (2021) that Post-local SGD does not improve much on ImageNet. For both stepwise decay and cosine decay learning rate schedules, the test accuracy improvement of Post-local SGD diminishes as learning rate decreases. Further, it remains unclear whether the generalization benefit continues to appear when the optimizer is changed from SGD to adaptive gradient methods such as Adam/AdamW, which are now indispensable for training large models.

**Our Contributions.**  In this paper, we aim to propose a general and effective $H$ schedule that can be readily applied to various optimizers and neural network models. Specifically, we introduce a simple yet effective strategy, called *Quadratic Synchronization Rule* (QSR), for dynamically adjusting the synchronization period according to the learning rate: given a learning rate schedule, we set $H$ proportional to $\eta^{-2}$ as the learning rate $\eta$ decays. This rule is largely inspired by a previous theoretical work (Gu et al., 2023), which shows that the generalization benefits arise only if $H = \Omega(\frac{1}{\eta})$ when $\eta \to 0$, but did not make any recommendation on how to set $H$.

Our main contributions are:

1. We propose the Quadratic Synchronization Rule (QSR) to simultaneously reduce the wall-clock time and improve the final test accuracy of local gradient methods. Based on the theoretical insights in Theorem 3.1, we provide a theoretical separation among data parallel SGD, Local SGD with $H \sim \eta^{-1}$, and Local SGD with QSR in terms of SDE approximations. We show that QSR can help reduce sharpness faster and hence improve generalization.

2. We demonstrate with ImageNet experiments that QSR can consistently improve the final test accuracy of ResNet-152 and ViT-B over other synchronization strategies, including constant-period and post-local schedules, and also $H \sim \eta^{-1}$ which one will expect to be optimal from the optimization perspective (Figure 1).

3. We thoroughly validate the efficacy of QSR not only for Local SGD but also for Local AdamW, which is arguably more suitable for training large models. We also validate its efficacy for cosine, linear and step decay learning rate schedules that are commonly used in practice.

4. We evaluate the communication efficiency of QSR on a 64-GPU NVIDIA GeForce RTX 3090 cluster. As an illustrative example, the standard data parallel AdamW takes 8.6 hours to train ViT-B for 300 epochs. With our QSR, Local AdamW cuts the training time down to 5.5 hours with even higher test accuracy.

## 2 OUR METHOD: QUADRATIC SYNCHRONIZATION RULE

Below we formulate the local gradient methods and present our Quadratic Synchronization Rule.

**Local Gradient Methods.** Given any gradient-based optimizer OPT, the corresponding local gradient method consists of multiple communication rounds. At the $s$-th round, each of the $K$ workers (say the $k$-th) gets a local copy of the global iterate $\bar{\boldsymbol{\theta}}^{(s)}$, i.e., $\boldsymbol{\theta}_{k,0}^{(s)} \leftarrow \bar{\boldsymbol{\theta}}^{(s)}$, and then performs $H$ steps of local updates. At the $h$-th local step of the $s$-th round, which corresponds to the $(sH+h)$-th iteration globally, each worker gets a batch of $B_{\mathrm{loc}}$ samples $(\xi_{k,h,1}^{(s)}, \ldots, \xi_{k,h,B_{\mathrm{loc}}}^{(s)})$ from a globally shared dataset $\tilde{D}$, computes the gradient on that batch, and updates the model with optimizer OPT and learning rate $\eta_{sH+h}$:

$$\boldsymbol{\theta}_{k,h+1}^{(s)} \leftarrow \mathrm{OPT}(\boldsymbol{\theta}_{k,h}^{(s)}, \eta_{sH+h}, \boldsymbol{g}_{k,h}^{(s)}) \quad \text{where} \quad \boldsymbol{g}_{k,h}^{(s)} = \frac{1}{B_{\mathrm{loc}}} \sum_{i=1}^{B_{\mathrm{loc}}} \nabla\ell(\boldsymbol{\theta}_{k,h}^{(s)}; \xi_{k,h,i}^{(s)}). \tag{1}$$

After finishing $H$ steps of local updates, all workers average their local models to generate the next global iterate: $\bar{\boldsymbol{\theta}}^{(s+1)} \leftarrow \frac{1}{K} \sum_{k=1}^{K} \boldsymbol{\theta}_{k,H}^{(s)}$. Note that conventional local gradient methods set the synchronization period as a constant, denoted as $H$, throughout training. See also Algorithm 2.

**Quadratic Synchronization Rule.** Given a learning rate schedule $\eta_t, t \in \{0, \cdots, T-1\}$ that decays with time, instead of keeping $H$ constant, we propose to dynamically increase the synchronization period $H^{(s)}$ at each round $s$ as the learning rate decreases. More specifically, if at the global iteration $t$ we need to start a new communication round, then we set

$$H^{(s)} := \max\left\{H_{\mathrm{base}}, \left\lfloor \left(\frac{\alpha}{\eta_t}\right)^2 \right\rfloor\right\}. \tag{2}$$

Here $H_{\mathrm{base}}$ is a constant indicating the minimum number of local steps one would like to use for each round, which should be set according to the relative cost of computation and communication. The coefficient $\alpha$, termed the "growth coefficient" henceforth, is a hyperparameter controlling how fast $H^{(s)}$ increases as $\eta_t$ decreases.

As suggested by our later theorem 3.1, $\alpha$ should be set as a small constant. In our experiments, we tune $\alpha$ properly between 0.01 and 0.5 and test the effectiveness of QSR with $H_{\mathrm{base}} = 2, 4, 8$. Note that the last communication round may not finish exactly at the last iteration of the learning rate schedule. If this is the case, we force a synchronization at the last step by setting $H^{(s)} := T - t$.

A surprising part of our method is that we use the power 2 in the above formula (2). This choice of power 2 is inspired by the analysis in Gu et al. (2023), which suggests that setting $H = \Omega(\frac{1}{\eta})$ is beneficial for reducing the sharpness of the local landscape. Indeed, $H^{(s)}$ could have been set to $H^{(s)} := \max\left\{H_{\mathrm{base}}, \left\lfloor \left(\frac{\alpha}{\eta_t}\right)^\gamma \right\rfloor\right\}$ for any $\gamma$. However, using $\gamma = 2$ is crucial for the success of our method, and we will provide theoretical justification and empirical evidence for this choice in Section 3. We also visualize the $H$ schedule for QSR in Figure 5 in the appendix.

**Dealing with Learning Rate Warmup.** Many learning rate schedules use a warmup phase where the learning rate increases linearly from 0 to $\eta_{\max}$, and then decays monotonically. This warmup phase is often used to avoid the instability caused by the initial large learning rate (Goyal et al., 2017). Our rule is not directly compatible with the warmup phase, since it is designed for a decaying learning rate, but the learning rate increases rather than decreases in this phase. Practically, we recommend setting $H^{(s)}$ as the value to be used in the communication round right after the warmup.

## 3 THEORETICAL MOTIVATIONS OF QUADRATIC SYNCHRONIZATION RULE

To justify our choice of power 2, we build on the same theoretical setup as Gu et al. (2023) to analyze the Stochastic Differential Equation (SDE) approximation of SGD and Local SGD using different scalings of $H$ with respect to $\eta$. Though the learning rate continuously decays over time in most of our experiments, it does not usually change much within a couple of epochs. Inspired by this, we take a quasistatic viewpoint: consider a significant period of time where the learning rate is relatively constant, and directly treat the learning rate as a real constant $\eta$. First, we recap Gu et al. (2023)'s theory that applies to Local SGD with $H \sim \eta^{-1}$, then we show how to generalize the result to our rule where $H \sim \eta^{-2}$, leading to a stronger implicit bias towards flatter minima.

**Setup.** Consider optimizing the loss function $\mathcal{L}(\boldsymbol{\theta}) := \mathbb{E}_{\xi \sim \tilde{\mathcal{D}}}[\ell(\boldsymbol{\theta}; \xi)]$, where $\boldsymbol{\theta} \in \mathbb{R}^d$ is the parameter vector and $\ell(\boldsymbol{\theta}; \xi)$ is the loss function for a single data sample $\xi$ drawn from a training set/training distribution $\tilde{\mathcal{D}}$. We use $\boldsymbol{\Sigma}(\boldsymbol{\theta}) := \mathrm{Cov}_{\xi \sim \tilde{\mathcal{D}}}[\nabla \ell(\boldsymbol{\theta}; \xi)]$ to denote the covariance matrix of the stochastic gradient $\nabla \ell(\boldsymbol{\theta}; \xi)$ at $\boldsymbol{\theta}$. Following Gu et al. (2023), we make regularity assumptions on $\mathcal{L}(\boldsymbol{\theta}), \boldsymbol{\Sigma}(\boldsymbol{\theta})$ and $\|\nabla \ell(\boldsymbol{\theta}; \xi)\|_2$ in Assumption E.1, and we assume that $\mathcal{L}$ has a manifold $\Gamma$ of minimizers in Assumption E.2. Our analysis is based on SDE approximations near $\Gamma$, providing a clean view of how different choices of $H$ affect the selection of minimizers by Local SGD.

**SDE approximations of SGD and Local SGD.** SDE is a powerful tool to precisely characterize the effect of noise in SGD, leading to many applications such as Linear Scaling Rule (Goyal et al., 2017). The SDE $\mathrm{d}\boldsymbol{\theta}(t) = -\nabla \mathcal{L}(\boldsymbol{\theta}(t))\mathrm{d}t + \frac{1}{\sqrt{B}}\boldsymbol{\Sigma}(\boldsymbol{\theta}(t))^{1/2}\mathrm{d}\boldsymbol{W}_t$ is conventionally used in the literature (Jastrzębski et al., 2017; Smith et al., 2020; Li et al., 2021b), where $\boldsymbol{W}_t$ is the standard Wiener process. In this SDE, each discrete step corresponds to a continuous time interval of length $\eta$, and the expected gradient and gradient noise become a deterministic drift term and a stochastic diffusion term, respectively. When the training proceeds to a point $\boldsymbol{\theta}(t)$ near a minimizer $\boldsymbol{\zeta}_0$ on the manifold $\Gamma$, the gradient $\nabla \mathcal{L}(\boldsymbol{\theta}(t))$ is almost zero but the gradient noise $\frac{1}{\sqrt{B}}\boldsymbol{\Sigma}(\boldsymbol{\theta}(t))^{1/2}\mathrm{d}\boldsymbol{W}_t$ drives the parameter to diffuse locally. This can be captured by a careful first-order approximation of the dynamics, leading to an Ornstein-Uhlenbeck process (Zhu et al., 2019; Li et al., 2019a; Izmailov et al., 2018). However, these rough approximations only hold for about $\mathcal{O}(\eta^{-1})$ steps, whereas neural networks in practice are usually trained for much longer.

Recently, a series of works (Blanc et al., 2020; Damian et al., 2021; Li et al., 2021c) study the dynamics of SGD on a *longer* horizon. They show that higher-order terms can accumulate over time and drive this local diffusion to gradually move on the manifold $\Gamma$. Among them, Li et al. (2021c) precisely characterized this with an SDE tracking the *gradient flow projection* of $\boldsymbol{\theta}(t)$ on $\Gamma$, denoted as $\Phi(\boldsymbol{\theta}(t))$ (see Definition E.1). Here, $\Phi(\boldsymbol{\theta}(t))$ can be thought of as a natural "center" of the local diffusion. This SDE, termed as Slow SDE, tracks the dynamics of SGD over $\mathcal{O}(\eta^{-2})$ steps, which is much longer than the $\mathcal{O}(\eta^{-1})$ horizon for conventional SDEs.

To provide a theoretical understanding of why Local SGD generalizes better than SGD, Gu et al. (2023) derived the Slow SDEs for Local SGD using the scaling $H \sim \eta^{-1}$. By comparing the Slow SDEs, they argued that Local SGD drifts faster to flatter minima than SGD. However, their analysis does not encompass the more aggressive scaling $H \sim \eta^{-2}$ recommended by our QSR. Recognizing this gap, we derive the Slow SDE for this scaling, enriching the theoretical framework for the generalization behavior of Local SGD. Below, we first present the Slow SDEs for SGD and Local SGD with $H \sim \eta^{-1}$ and $H \sim \eta^{-2}$, then we interpret why $H \sim \eta^{-2}$ may generalize better.

**Definition 3.1** (Slow SDE for SGD, informal, (Li et al., 2021c; Gu et al., 2023))**.** *Given* $\boldsymbol{\zeta}_0 \in \Gamma$, *define* $\boldsymbol{\zeta}(t)$ *as the solution to the following SDE with initial condition* $\boldsymbol{\zeta}(0) = \boldsymbol{\zeta}_0$:

$$\mathrm{d}\boldsymbol{\zeta}(t) = P_{\boldsymbol{\zeta}}\bigg(\underbrace{\frac{1}{\sqrt{B}}\boldsymbol{\Sigma}_{\parallel}^{1/2}(\boldsymbol{\zeta})\mathrm{d}\boldsymbol{W}_t}_{\text{(a) diffusion on } \Gamma} \underbrace{-\frac{1}{2B}\nabla^3\mathcal{L}(\boldsymbol{\zeta})[\widehat{\boldsymbol{\Sigma}}_{\Diamond}(\boldsymbol{\zeta})]\mathrm{d}t}_{\text{(b) drift on } \Gamma}\bigg). \tag{3}$$

*Here, $P_\zeta$ is a projection operator of differential forms to ensure that taking an infinitesimal step from $\zeta \in \Gamma$ remains on the manifold $\Gamma$. $B$ is the total batch size. $\Sigma_\|(\zeta)$ and $\widehat{\Sigma}_\diamond(\zeta)$ are certain PSD matrices related to gradient noise and Hessian. See Definition E.2 for the full definition.*

**Definition 3.2** (Slow SDE for Local SGD with $H \sim \eta^{-1}$, informal (Gu et al., 2023))**.** *Consider the scaling $H = \beta/\eta$ for some constant $\beta$. Given $\zeta_0 \in \Gamma$, define $\zeta(t)$ as the solution to the following SDE with initial condition $\zeta(0) = \zeta_0$:*

$$\mathrm{d}\zeta(t) = P_\zeta \Big( \underbrace{\tfrac{1}{\sqrt{B}}\Sigma_\|^{1/2}(\zeta)\mathrm{d}W_t}_{\textit{(a) diffusion on }\Gamma} \underbrace{-\tfrac{1}{2B}\nabla^3\mathcal{L}(\zeta)[\widehat{\Sigma}_\diamond(\zeta)]\mathrm{d}t}_{\textit{(b) drift on }\Gamma\textit{, same as SGD}} \underbrace{-\tfrac{K-1}{2B}\nabla^3\mathcal{L}(\zeta)[\widehat{\Psi}(\zeta; H\eta)]\mathrm{d}t}_{\textit{(c) an extra drift term on }\Gamma} \Big), \quad (4)$$

*where $K$ is the number of workers, $B, \Sigma_\|(\zeta)$ and $\widehat{\Sigma}_\diamond(\zeta)$ are the same as in Definition 3.1. Here, $\widehat{\Psi}(\zeta; \beta)$ is a PSD matrix depending on gradient noise and Hessian. It scales with $\beta$ as $\lim_{\beta\to 0} \widehat{\Psi}(\zeta; \beta) = 0$, $\lim_{\beta\to+\infty} \widehat{\Psi}(\zeta; \beta) = \widehat{\Sigma}_\diamond(\zeta)$. [1] See Definition E.3 for the full definition.*

**Definition 3.3** (Slow SDE for Local SGD with QSR)**.** *Given $\zeta_0 \in \Gamma$, define $\zeta(t)$ as the solution to the following SDE with initial condition $\zeta(0) = \zeta_0$:*

$$\mathrm{d}\zeta(t) = P_\zeta \Big( \underbrace{\tfrac{1}{\sqrt{B}}\Sigma_\|^{1/2}(\zeta)\mathrm{d}W_t}_{\textit{(a) diffusion on }\Gamma} \underbrace{-\tfrac{K}{2B}\nabla^3\mathcal{L}(\zeta)[\widehat{\Sigma}_\diamond(\zeta)]\mathrm{d}t}_{\textit{(b) drift on }\Gamma\textit{, }K\textit{ times larger}} \Big), \quad (5)$$

*where $K, B, \Sigma_\|(\zeta)$ and $\widehat{\Sigma}_\diamond(\zeta)$ are defined in Definitions 3.1 and 3.2.*

The following approximation theorem indicates that when the learning rate $\eta$ and the growth coefficient $\alpha$ for QSR are small, the above Slow SDEs closely track their discrete counterparts. The approximation theorem for QSR is new, and we defer the proof to Appendix E.2.

**Theorem 3.1** (Weak Approximations)**.** *Let $T > 0$ be a constant and $\zeta(t)$ be the solution to one of the above Slow SDEs with the initial condition $\zeta(0) = \Phi(\theta^{(0)}) \in \Gamma$. Let $g(\theta)$ be any $\mathcal{C}^4$-smooth function.*

1. *(Gu et al., 2023) For SGD, let $\zeta(t)$ be the solution to (3). Then, $\max_{0 \le s \le \frac{T}{\eta^2}} |\mathbb{E}[g(\Phi(\theta_s))] - \mathbb{E}[g(\zeta(s\eta^2))]| = \tilde{\mathcal{O}}(\eta^{0.25})$.*

2. *(Gu et al., 2023) For Local SGD with $H = \beta/\eta$ for some constant $\beta$, let $\zeta(t)$ be the solution to (4). Then, $\max_{0 \le s \le \frac{T}{H\eta^2}} |\mathbb{E}[g(\Phi(\theta^{(s)}))] - \mathbb{E}[g(\zeta(sH\eta^2))]| = \tilde{\mathcal{O}}(\eta^{0.25})$.*

3. *For Local SGD with $H = (\frac{\alpha}{\eta})^2$, where the positive constant $\alpha$ is small but larger than $\Omega(\eta^\gamma)$ for all $\gamma > 0$, let $\zeta(t)$ be the solution to (5). Then, $\max_{0 \le s \le \frac{T}{H\eta^2}} |\mathbb{E}[g(\Phi(\theta^{(s)}))] - \mathbb{E}[g(\zeta(sH\eta^2))]| = \mathcal{O}(\alpha^2)$.*

*Here, $\mathcal{O}(\cdot)$ and $\tilde{\mathcal{O}}(\cdot)$ hide constants that are independent of $\alpha$ and $\eta$ but can depend on $g$ and $T$. $\tilde{\mathcal{O}}(\cdot)$ also hides log terms.*

By comparing the Slow SDEs, we can predict the generalization order for different scaling as **QSR** $>$ **{H $\sim \eta^{-1}$}** $>$ **{constant H}**, which we explain in detail below.

**Interpretation of the Slow SDEs.** We first focus on the Slow SDE for SGD (3). The key component of this Slow SDE is the drift term (b), which comes from higher-order approximations of the aforementioned local diffusion that happens in $\mathcal{O}(\eta^{-1})$ steps. Viewing $\nabla^3\mathcal{L}(\zeta)[\widehat{\Sigma}_\diamond(\zeta)]$ as a semi-gradient of $\langle\nabla^2\mathcal{L}(\zeta), \widehat{\Sigma}_\diamond(\zeta)\rangle$ that discards the dependence of $\theta$ in $\widehat{\Sigma}_\diamond(\zeta)$, we can interpret the Slow SDE as a continuous version of a semi-gradient method for reducing $\langle\nabla^2\mathcal{L}(\zeta), \widehat{\Sigma}_\diamond(\zeta)\rangle$ on $\Gamma$. Since the Hessian matrix $\nabla^2\mathcal{L}(\zeta)$ determines the local curvature of the loss landscape, we can conclude from the Slow SDE that SGD tends to reduce sharpness and move towards flatter minimizers in $\mathcal{O}(\eta^{-2})$ steps. Reduced sharpness has been shown to yield better sample complexity bounds in specific theoretical settings. For details, we refer the readers to Li et al. (2021c).

---

[1]Given $\zeta$, $\widehat{\Psi}(\zeta; \beta)$ is a monotonically increasing function of $\beta$ in the eigenspace of the Hessian matrix $\nabla^2\mathcal{L}(\zeta)$.

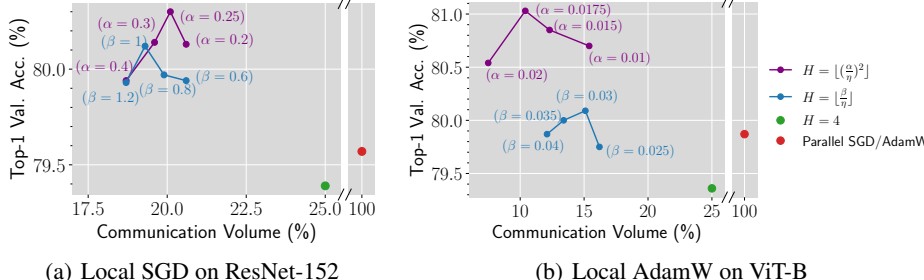

(a) Local SGD on ResNet-152      (b) Local AdamW on ViT-B

Figure 2: Empirical results on Local SGD and Local AdamW validate the generalization performance order predicted by our theory: QSR > $\{H \sim \eta^{-1}\}$ > {constant $H$}. For SGD, we additionally have {constant $H$} $\approx$ {parallel SGD} since the latter is equivalent to Local SGD with $H = 1$. Here, $\alpha$ and $\beta$ are tuned to maximize the test accuracy of QSR and $H \sim \eta^{-1}$, respectively.

Now, we turn to the Slow SDE for QSR. Compared with the SDE for SGD, it possesses a $K$ times larger drift term, leading to much faster sharpness reduction than SGD. An intuitive explanation for why this extra drift arises is as follows. Since the local batch size is $K$ times smaller than the global one, this local diffusion at each worker is much more significant than that in parallel SGD, thereby leading to an extra drift term in Slow SDE accumulated from higher-order terms.

The case of Local SGD with $H = \beta/\eta$ is somewhere in between QSR and SGD. Compared with the SDE for SGD, it has an extra drift term (c), where $\beta$ serves as the knob to control the magnitude of the drift term. For small $\beta$, $\widehat{\mathbf{\Psi}}(\zeta)$ diminishes to zero, yielding the same SDE as SGD. By contrast, as $\beta$ goes to infinity, $\widehat{\mathbf{\Psi}}(\zeta)$ approximates $\widehat{\mathbf{\Sigma}}_\Diamond(\zeta)$, leading to the Slow SDE for QSR.

**Comparison of different scalings.** Based on the interpretation, keeping $H$ constant as $\eta$ diminishes is equivalent to setting a small $\beta$ for $H = \beta/\eta$, making the extra drift term negligible and thus yielding nearly no generalization benefit over SGD. Conversely, the SDE for $H = \beta/\eta$ converges to the SDE of QSR in the limit $\beta \to \infty$, maximizing the drift term. But in practice, $\beta$ cannot be arbitrarily large. In Theorem 3.3 of Gu et al. (2023), the distance between the iterate and $\Gamma$ blows up as $\tilde{\mathcal{O}}(\sqrt{\beta\eta})$, suggesting that setting a very large $\beta$ for a not-so-small $\eta$ can blow up the loss. Therefore, the generalization performance of $H \sim \eta^{-1}$ is expected to be worse than QSR. In summary, the order of generalization performance predicted by our theory is QSR > $\{H \sim \eta^{-1}\}$ > {constant $H$}.

Experimental results in Figure 2 validate that this order of generalization performance for different scalings holds not only for Local SGD but also for Local AdamW. For Local SGD we additionally have {constant $H$} $\approx$ {parallel SGD} since parallel SGD is mathematically equivalent to Local SGD with $H = 1$. Apart from $H \sim \eta^{-1}$ and $H \sim \eta^{-2}$, we also tried a more aggressive scaling, $H \sim \eta^{-3}$, but it does not provide consistent improvements over QSR. See Appendix G for more discussion.

## 4 EXPERIMENTS

In this section, we empirically demonstrate that QSR not only improves the test accuracy of local gradient methods but also reduces the wall-clock time of standard data parallel training, with a focus on the ImageNet classification task (Russakovsky et al., 2015). Our experiments include Local SGD on ResNet-152 (He et al., 2016), and Local AdamW on ViT-B with patch size 16x16 (Dosovitskiy et al., 2021). We briefly outline our training configuration below. See Appendix C for full details.

**Baselines.** For QSR with base synchronization period $H_{\text{base}}$, we benchmark their performance against two baselines running the same number of epochs: ① Local SGD/AdamW with constant synchronization period $H = H_{\text{base}}$, and ② parallel SGD/AdamW. When comparing with these baselines, we mainly focus on validating that (a) QSR maintains or sometimes outperforms the communication efficiency of ①, thus communicating much less than ②, and (b) QSR improves the generalization performance of ①, even surpassing ② in test accuracy.

**Comparison with other synchronization strategies.** Besides the above two baselines, other potential baselines include ③ Post-local SGD, ④ the scaling of $H \sim \eta^{-1}$, and ⑤ large batch training with batch size $H \times B$, which we discuss below. ③ is proposed for the same purpose as QSR: to improve communication efficiency and generalization together. However, it is less communication efficient than our QSR because it starts with parallel SGD and sustains this for a significant fraction of the training duration, leading to a limited reduction in communication. Also, as shown by

Table 1: QSR enhances the test accuracy of local gradient methods, even outperforming the communication-intensive data parallel approach. The experiments below use batch size 4096. We report the validation accuracy and train loss averaged over 3 runs, along with the standard deviation.

| (a) Local SGD on ResNet-152 | | | |
| --- | --- | --- | --- |
| Method | Val. acc. (%) | Train loss | Comm. |
| Parallel SGD | 79.53 (0.07) | 1.57 (0.01) | 100% |
| Local SGD ($H$=2) | 79.54 (0.07) | **1.58** (0.00) | 50% |
| + QSR ($H_{\text{base}}$=2) | **80.30** (0.04) | 1.67 (0.01) | **39.7%** |
| Local SGD ($H$=4) | 79.48 (0.12) | **1.62** (0.02) | 25% |
| + QSR ($H_{\text{base}}$=4) | **80.27** (0.05) | 1.69 (0.01) | **20.1%** |

| (b) Local AdamW on ViT-B | | | |
| --- | --- | --- | --- |
| Method | Val. acc. (%) | Train loss | Comm. |
| Parallel AdamW | 79.86 (0.03) | 1.09 (0.00) | 100% |
| Local AdamW ($H$=4) | 79.32 (0.06) | **1.01** (0.02) | 25% |
| + QSR ($H_{\text{base}}$=4) | **80.98** (0.05) | 1.32 (0.00) | **10.4%** |
| Local AdamW ($H$=8) | 78.93 (0.10) | **1.06** (0.00) | 12.5% |
| + QSR ($H_{\text{base}}$=8) | **80.56** (0.10) | 1.35 (0.01) | **6.9%** |

our comparison in Figure 1(a) (also observed in Ortiz et al. 2021), its generalization benefits over SGD appear shortly after switching and diminish in the end. ④ is inspired by Gu et al. (2023) and may also improve generalization while reducing communication, but we have conducted a thorough comparison between QSR and ④ in Figure 2, demonstrating the superiority of QSR. ⑤ has the same communication efficiency as Local SGD with the same constant $H$ (①), but it has been observed to have worse test accuracy than parallel SGD/AdamW without scaling up the batch size (②), which we also observe in Table 2. For the above reasons, we mainly compare with baselines ① and ②.

**Hardware.** We conduct the experiments on Tencent Cloud, where each machine is equipped with 8 NVIDIA GeForce RTX 3090 GPUs. The machines are interconnected by a 25Gbps network. Since intra-machine communication speed is not substantially faster than inter-machine speed on our specific hardware, we treat *each GPU* as an independent worker and set the batch size on each GPU as $B_{\text{loc}} = 256$. In this paper, we use $a$x$b$ GPUs to denote $a$ machines with $b$ GPUs each.

**Training Setup.** Our experiments on ResNet-152 follow the 200-epoch recipe in Foret et al. (2021b) except that we use 5 epochs of linear learning rate warmup. For experiments on ViT-B, we follow the simple and effective 300-epoch recipe proposed in Beyer et al. (2022) with RandAugment and Mixup. We use the cosine decay unless otherwise stated. The hyperparameters (primarily learning rate and weight decay) are optimally tuned for all baselines. We explore $H_{\text{base}} = 2, 4$ for ResNet-152 and $H_{\text{base}} = 4, 8$ for ViT-B. This choice stems from the observation that the communication overhead for ResNet-152 is smaller than ViT-B (see Table 4). To tune the growth coefficient $\alpha$ for QSR, we first fix the learning rate schedule and then search among a few values of $\alpha$. The $\alpha$ values we explore typically allow the training to start with $H_{\text{base}}$, maintain $H = H_{\text{base}}$ for an initial period to optimize the training loss, and gradually increase $H$ as $\eta$ decays in the late phase.

### 4.1 QSR IMPROVES GENERALIZATION

Through experiments spanning various batch sizes and learning rate schedules, in this subsection, we illustrate that QSR consistently enhances the generalization of gradient methods, even outperforming the communication-intensive data parallel approach.

**Main results.** We first present our main results for batch size $B = 4096$ on 2x8 GPUs, covering Local SGD on ResNet-152 and Local AdamW on ViT-B. As shown in Table 1, QSR significantly improves the validation accuracy of local gradient methods by up to $0.8\%$ on ResNet-152 and $1.7\%$ on ViT-B, despite inducing higher training loss. The results support the thesis that the improvement in generalization is due to the implicit regularization of local gradient noise instead of better optimization. Noticeably, QSR surpasses the data parallel approach in validation accuracy by $0.7\%$ on ResNet-152 and by $1.1\%$ on ViT-B while cutting the communication volume to less than $25\%$. As an added benefit of increasing the synchronization interval in line with the decaying learning rate, QSR further reduces communication overhead, even halving the communication volume compared to Local AdamW with a fixed synchronization period on ViT-B.

The advantages of QSR are more pronounced for ViT-B compared to ResNet-152. This is probably because vision transformers are general-purpose architectures with less image-specific inductive bias than CNNs (Dosovitskiy et al., 2021; Chen et al., 2021). As a result, they may benefit more from external regularization effects, such as those induced by adding local steps.

**Scaling up the batch size.** In Table 2, when scaling the training up to 8x8 GPUs with total batch size $B = 16384$, we observe a drop in test accuracy for both data parallel approach and local gradient methods. This generalization degradation for large batch training, which has been widely observed

Table 2: QSR mitigates the generalization degradation in large-batch training. Here the batch size is 16384.

(a) Local SGD on ResNet-152

| Method | Val. Acc.(%) | Comm. (%) |
|---|---|---|
| Parallel SGD | 79.20 | 100 |
| Local SGD ($H$=2) + QSR ($H_{base} = 2$) | 78.67 **79.27** | 50 **42.8** |
| Local SGD ($H$=4) + QSR ($H_{base} = 4$) | 78.34 **78.65** | 25 **21.9** |

(b) Local AdamW on ViT-B

| Method | Val. Acc. (%) | Comm. (%) |
|---|---|---|
| Parallel AdamW | 78.52 | 100 |
| Local AdamW ($H$=4) +QSR ($H_{base} = 4$) | 77.83 **79.36** | 25 **16.1** |
| Local AdamW ($H$=8) +QSR ($H_{base} = 8$) | 77.62 **78.26** | 12.5 **9.8** |

Table 3: QSR also exhibits strong generalization performance on the step-decay learning rate schedule.

(a) Local SGD on ResNet-152.

| Method | Val. Acc. (%) | Comm. (%) |
|---|---|---|
| Parallel SGD | 79.68 | 100 |
| Local SGD ($H$=2) +QSR ($H_{base} = 2$) | 79.58 **80.40** | 50 **40.3** |
| Local SGD ($H$=4) +QSR ($H_{base} = 4$) | 79.53 **80.11** | 25 **20.5** |

(b) Local AdamW on ViT-B.

| Method | Val. Acc.(%) | Comm.(%) |
|---|---|---|
| Parallel AdamW | 79.91 | 100 |
| Local AdamW ($H$=4) + QSR ($H_{base} = 4$) | 79.36 **80.9** | 25 **12.7** |
| Local AdamW ($H$=8) + QSR($H_{base} = 8$) | 79.23 **80.65** | 12.5 **7.2** |

in the literature (Shallue et al., 2019; Jastrzębski et al., 2017; You et al., 2018), probably arises from a reduced level of gradient noise associated with increased batch size (Keskar et al., 2017b; Smith et al., 2021). While the Linear Scaling Rule for SGD (Krizhevsky, 2014; Goyal et al., 2017) and the Square Root Scaling Rule (Malladi et al., 2022; Granziol et al., 2022) for adaptive gradient methods – which increase the learning rate in proportion to the total batch size or its square root – can mitigate this degradation, they cannot fully bridge the gap. In Table 2, the test accuracy drop persists even when we tune the learning rate for all baselines. Applying QSR to local gradient methods can help reduce this generalization gap. It improves the validation accuracy of local gradient methods by up to $0.6\%$ on ResNet-152 and $1.5\%$ on ViT-B. This enables local gradient methods to achieve comparable validation accuracy as the data parallel approach on ResNet or outperform it by $0.8\%$ on ViT while communicating considerably less.

**Other learning rate schedules.** So far, our experiments are conducted with the cosine learning rate schedule, which is a common choice for training modern deep neural nets (Liu et al., 2021; 2022; Brown et al., 2020). To further validate the efficacy of QSR, we now investigate other popular learning rate schedules, including linear (Li et al., 2020a; Izsak et al., 2021; Leclerc et al., 2023) and step decay (He et al., 2016; Huang et al., 2017; Ma et al., 2019). See Figure 4 for a visualization of these schedules. Figure 3 presents the results for Local AdamW on ViT-B with linear decay, where the peak learning rates for baselines are tuned optimally. QSR improves the test accuracy of Local AdamW by a significant margin of $1.4\%$, even outperforming parallel AdamW by $0.6\%$ while cutting the communication volume to only $9.3\%$. The step

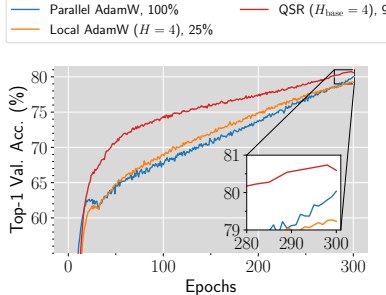

Figure 3: For linear decay, QSR improves the test accuracy of Local AdamW on ViT-B, even outperforming the communication-intensive parallel AdamW.

decay scheduler divides the learning rate by factors such as 2 or 10 at some specified epochs. Given the absence of standard recipes to determine the decay points in our training setup, we derive a step decay schedule from the cosine decay by rounding its learning rate to powers of 2, which is defined as $\eta_{step}(t) := 2^{round(\log_2 \eta_{cos}(t))}$. As shown in Table 3, QSR exhibits strong generalization performance with this decay schedule, enhancing the test accuracy of local gradient methods by up to $0.8\%$ on ResNet-152 and $1.5\%$ on ViT-B. It even surpasses the communication-intensive parallel SGD by $0.7\%$ on ResNet and parallel AdamW by $1\%$ on ViT.

## 4.2 QSR REDUCES WALL-CLOCK TIME

In addition to improving generalization, our original motivation for adopting local steps is to reduce communication overhead and hence reduce the wall-clock time. In this section, we confirm this for training with 2x8 and 8x8 GPUs, as shown in Table 4. See also Appendix F for our method of measuring the communication time. In our setup, scaling the training from 2x8 to 8x8 GPUs

Table 4: QSR reduces the wall-clock time of data parallel training. The following tables present wall-clock time for the entire training process on 2x8 GPUs and 8x8 GPUs, with batch sizes 4096 and 16384, respectively. We highlight the wall-clock time of QSR when it matches or outperforms the data parallel baseline in test accuracy. "Ratio" represents communication time divided by total time, reflecting the communication overhead. We also include local gradient methods with a constant synchronization period for reference.

(a) ResNet-152 (200 epochs) on 2x8 GPUs

| Method | Comm. (h) | Total (h) | Ratio (%) |
|---|---|---|---|
| Parallel SGD | 3.3 | 20.7 | 15.9 |
| QSR ($H_{base} = 2$) | 1.3 | **18.7** | 7.0 |
| QSR ($H_{base} = 4$) | 0.7 | **18.0** | 3.9 |
| Local SGD ($H$=2) | 1.6 | 19.0 | 8.4 |
| Local SGD ($H$=4) | 0.8 | 18.0 | 4.4 |

(b) ViT-B (300 epochs) on 2x8 GPUs

| Method | Comm. (h) | Total (h) | Ratio(%) |
|---|---|---|---|
| Parallel AdamW | 7.3 | 26.7 | 27.3 |
| QSR ($H_{base} = 4$) | 0.8 | **20.2** | 4.0 |
| QSR ($H_{base} = 8$) | 0.5 | **20.0** | 2.5 |
| Local AdamW ($H$=4) | 1.8 | 21.2 | 8.4 |
| Local AdamW ($H$=8) | 0.9 | 20.5 | 4.4 |

(c) ResNet-152 (200 epochs) on 8x8 GPUs

| Method | Comm. (h) | Total (h) | Ratio (%) |
|---|---|---|---|
| Parallel SGD | 1.3 | 5.7 | 22.8 |
| QSR ($H_{base} = 2$) | 0.6 | **5.0** | 12.0 |
| QSR ($H_{base} = 4$) | 0.3 | 4.7 | 6.4 |
| Local SGD ($H$=2) | 0.7 | 5.1 | 13.7 |
| Local SGD ($H$=4) | 0.3 | 4.8 | 6.3 |

(d) ViT-B (300 epochs) on 8x8 GPUs

| Method | Comm. (h) | Total (h) | Ratio (%) |
|---|---|---|---|
| Parallel AdamW | 3.7 | 8.6 | 43.0 |
| QSR ($H_{base} = 4$) | 0.6 | **5.5** | 10.9 |
| QSR ($H_{base} = 8$) | 0.4 | 5.3 | 7.5 |
| Local AdamW ($H$=4) | 0.9 | 5.8 | 15.5 |
| Local AdamW ($H$=8) | 0.5 | 5.3 | 9.4 |

increases the communication overhead for both models. Notably, on 8x8 GPUs, communication accounts for almost half of the total training time for ViT-B. Since communication makes up a larger portion of the total time for ViT-B compared to ResNet-152, the speedup from QSR is more significant on ViT-B: the time is cut from 26.7 to 20.2 hours on 2x8 GPUs, and 8.6 to 5.5 hours on 8x8 GPUs. As discussed in Section 4.1, compared to the constant period local gradient method, QSR further reduces the communication cost by increasing the synchronization period in the late phase. For example, applying QSR to Local AdamW with $H = 4$ further reduces the time by 1 hour for ViT training on 2x8 GPUs.

**Discussion on the choice of $H_{base}$.** As elaborated in Section 2, $H_{base}$ indicates the minimum synchronization period and should be determined based on the communication overhead. For ResNet-152, given that communication only accounts for 3.3 out of 20.7 hours on 2x8 GPUs and 1.3 out of 5.7 hours on 8x8 GPUs, setting $H_{base}$ as 2 or 4 suffices to reduce the communication time to an inconsequential amount. By contrast, the communication overhead for ViT-B is more prominent, motivating us to consider larger values of $H_{base}$, such as 4 and 8. As shown in Tables 1 and 2, $H_{base}$ introduces a tradeoff between communication efficiency and final test accuracy. For instance, when training ResNet-152 with batch size 16384, one can either choose $H_{base} = 2$ to achieve comparable test accuracy as parallel SGD, or $H_{base} = 4$ to further halve the communication volume at the expense of a $0.6\%$ drop in test accuracy. One probable explanation for this accuracy drop for larger $H_{base}$ can be worse optimization in the early training phase, where the learning rate is large.

## 5 DISCUSSIONS AND FUTURE DIRECTIONS

This paper primarily focuses on relatively large models trained with long horizons, and proposes the Quadratic Synchronization Rule (QSR). As validated by our experiments, QSR effectively improves test accuracy and communication efficiency simultaneously for training large vision models (ResNet-152 and ViT-B) with quite a few hundred epochs. However, on the downside, for smaller models trained with shorter horizons, QSR may not consistently deliver noticeable generalization improvements (see Table 5). Nonetheless, training in this regime is not costly, either, making it less of a critical concern. Another limitation of our work is that the effectiveness of QSR relies on the implicit regularization effects of noise, but regularization techniques become less important in bridging the gap between the training and population loss (Vyas et al., 2023) in pertaining large model with unsupervised learning, where the training is done on massive data with only a few epochs. Still, certain implicit/explicit regularization effects have been found to be effective in improving downstream performance despite the same pertaining loss (Liu et al., 2023; Panigrahi et al., 2024). We leave it to future work to explore and design communication-efficient methods for unsupervised learning, particularly language model pretraining, that improve models' transferability to downstream tasks.

ACKNOWLEDGEMENT AND DISCLOSURE OF FUNDING

The work of Xinran Gu and Longbo Huang is supported by the Technology and Innovation Major Project of the Ministry of Science and Technology of China under Grant 2020AAA0108400 and 2020AAA0108403. The work of Kaifeng Lyu and Sanjeev Arora is partly supported by NSF and ONR.

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

## CONTENTS

## A  ADDITIONAL RELATED WORKS

**Advances in local gradient methods.**  Local gradient methods are a class of communication-efficient algorithms for distributed training. In this approach, workers update their models locally and average the model parameters every time they finish $H$ steps of updates. Dating back to Mann et al. (2009) and Zinkevich et al. (2010), local gradient methods have been widely used to improve communication efficiency in both datacenter distributed training Zhang et al. (2014); Povey et al. (2014); Su & Chen (2015); Chen & Huo (2016) and Federated Learning (Kairouz et al., 2021; McMahan et al., 2017; Li et al., 2019b; Konečnỳ et al., 2016). Many variants have been proposed to facilitate the convergence speed. Examples include using control variates (Karimireddy et al., 2020), adding proximal terms to local loss functions (Li et al., 2020b), and applying adaptivity on top of each communication round (Wang et al., 2019; Reddi et al., 2020). Local gradient methods can also be readily combined with orthogonal approaches like communication compression (Basu et al., 2019) and asynchronous updates (Nadiradze et al., 2021) for further communication cost reduction.

**Optimization perspectives on selecting $H$.**  Extensive prior research has been devoted to optimizing the selection of the synchronization period $H$ from an optimization perspective. The conventional approach sets $H$ as a constant throughout training. In this setup, a series of studies (e.g.,Khaled et al. (2020); Stich (2018); Haddadpour et al. (2019); Yu et al. (2019)) established convergence bounds for the training loss, which typically degrade as $H$ gets larger. leading to a trade-off between communication efficiency and model accuracy. Drawing upon these theoretical results, $H$ should be set as the smallest value that reduces the communication cost to an acceptable level to minimize the negative impact on optimization. To better trade-off between optimization and generalization, researchers introduced various adaptive communication strategies. Kamp et al. (2014) designed a synchronization protocol controlled by the variance in model parameters. Haddadpour et al. (2019) suggested linearly increasing $H$ as the iteration goes on. Shen et al. (2021) introduced a stagewise communication scheme that halves the learning rate $\eta$ while doubles $H$ every time the training has finished a predefined stage. Aimed at optimizing the convergence of training loss with respect to wall-clock time, Wang & Joshi (2019) proposed a strategy that starts with infrequent communication and gradually decreases $H$ as training progresses. Nonetheless, the effectiveness of these adaptive communication strategies has only been empirically validated on linear models or small-scale datasets like CIFAR-10/100.

**Generalization perspectives on selecting $H$.**  While a larger $H$ usually hurts optimization, it can sometimes improve generalization. Apart from Lin et al. (2020) that has been discussed in detail in Section 1, similar observations have been reported by Gupta et al. (2020) and Wortsman et al. (2023). Specifically, Gupta et al. (2020) introduced the Stochastic Weight Averaging in Parallel (SWAP) algorithm, which runs parallel SGD until a target training accuracy, then lets workers perform local updates with a final model averaging. Their empirical results validate SWAP's superior generalization performance over parallel SGD. When using LAMB (You et al., 2020) as the optimizer, Wortsman et al. (2023) find that complete local fine-tuning, followed by a single model averaging in the end (equivalent to setting $H$ as the total number of iterations), outperforms the standard parallel LAMB in test accuracy under distribution shifts. Another relevant method is the "model soup" (Wortsman et al., 2022), which averages multiple models fine-tuned with different hyperparameters and turns out to beat the single model in test accuracy. Our paper focuses on designing the synchronization scheme best for generalization.

**Implicit bias of optimizers.**  The success of deep learning lies in its remarkable ability to generalize to unseen data, though it possesses the capacity to fit randomly labeled data (Zhang et al., 2017). A significant contributing factor to this success is the implicit bias inherent in popular optimizers like Gradient Descent (GD) and Stochastic Gradient Descent (SGD). Specifically, these optimizers favor minima that exhibit good generalization, without explicitly encoding such bias into the training loss. A lot of studies have been devoted to characterizing this implicit bias, some through the lens of margin maximization (Soudry et al., 2018b;a; Lyu & Li, 2020; Ji & Telgarsky, 2020; Chizat & Bach, 2020; Nacson et al., 2019), and some others focus on the simplicity bias from small initialization (Li et al., 2018; Razin & Cohen, 2020; Arora et al., 2019; Li et al., 2021a; Lyu et al., 2021; Razin et al., 2022; Stöger & Soltanolkotabi, 2021; Ge et al., 2021; Jin et al., 2023). The line of work most closely related to our paper interprets the implicit bias via sharpness reduction. The connection between flatter minima and better generalization is a commonly held belief that has

been investigated both theoretically (Hochreiter & Schmidhuber, 1997; Neyshabur et al., 2017) and empirically (Keskar et al., 2017a; Jiang et al., 2020). Drawing on this insight, Foret et al. (2021a) introduced SAM optimizer, which delivers superior generalization performance by explicitly penalizing sharpness. Recent theoretical studies (Arora et al., 2022; Lyu et al., 2022; Damian et al., 2023; Ma et al., 2022) elucidate that GD inherently biases towards flatter regions on the loss landscape. Specifically, under some regularity conditions, they show that GD will eventually enter the "Edge of Stability"(Cohen et al., 2020), where the maximum eigenvalue of the loss Hessian stays around 2/learning rate, and then constantly moves towards flatter minima. Going beyond GD, another line of work studies how gradient noise in SGD helps reduce sharpness. Wu et al. (2018); Hu et al. (2017); Ma & Ying (2021) showed that gradient noise can cause training instability around sharp minima, and hence, the iterate can only settle around flat minima. Kleinberg et al. (2018); Zhu et al. (2019); Xie et al. (2021); Ibayashi & Imaizumi (2021) analyzed the escaping behavior of SGD from sharp minima. Motivated by recent empirical observations that low-loss solutions on the loss landscape are path-connected (Garipov et al., 2018; Draxler et al., 2018; Frankle et al., 2020) rather than isolated, Blanc et al. (2020); Damian et al. (2021); Li et al. (2021c) assume the existence of a minimizer manifold and show that gradient noise provably drives the iterate towards flatter minima on this manifold. Cowsik et al. (2022); Wang et al. (2023) discuss how momentum preserves or strengthens this effect. Also through the lens of sharpness reduction, the recent work by Gu et al. (2023) explains the generalization benefit of Local SGD, as discussed in Section 3. Zhu et al. (2023) elucidate that a similar implicit bias also manifests in decentralized training by making connections to certain variants of SAM.

## B  PSEUDOCODE

We present the pseudocodes for standard data parallel methods and local gradient methods below.

---

**Algorithm 1:** `Parallel OPT`: Data Parallel Methods on $K$ Workers

---

1 **Input**: loss function $\ell(\boldsymbol{\theta}; \xi)$, initial parameter $\boldsymbol{\theta}^{(0)}$
2 **Hyperparameters**: total number of iterations $T$
3 **Hyperparameters**: learning rate schedule $\eta_t, t \in \{0, \cdots, T\}$, local batch size $B_{\mathrm{loc}}$

4 $t \leftarrow 0$ ;             // initialize the global iteration number
5 **for** $t = 0, \ldots, R - 1$ **do**
6     **for** *each worker* $k$ **do in parallel**
7        $(\xi_{k,t,1}^{(s)}, \ldots, \xi_{k,t,B_{\mathrm{loc}}}^{(s)}) \leftarrow$ `Sample()` ;      // sample a local batch
8        $\boldsymbol{g}_k^{(t)} \leftarrow \frac{1}{B_{\mathrm{loc}}} \sum_{i=1}^{B_{\mathrm{loc}}} \nabla \ell(\boldsymbol{\theta}^{(t)}; \xi_{k,i}^{(t)})$ ;      // computing the local gradient
9     **end**
10     $\boldsymbol{g}^{(t)} \leftarrow \frac{1}{K} \sum_{k=1}^{K} \boldsymbol{g}_k^{(t)}$ ;      // All-Reduce aggregation of local gradients
11     $\boldsymbol{\theta}^{(t+1)} \leftarrow \mathrm{OPT}(\boldsymbol{\theta}^{(t)}, \eta_t, \boldsymbol{g}^{(t)})$ ;      // update the model with optimizer OPT
12 **end**

---

**Algorithm 2:** `Local OPT`: Local Gradient Methods on $K$ Workers

---

1 **Input**: loss function $\ell(\boldsymbol{\theta}; \xi)$, initial parameter $\bar{\boldsymbol{\theta}}^{(0)}$
2 **Hyperparameters**: total number of rounds $R$
3 **Hyperparameters**: learning rate schedule $\eta_t, t \in \{0, \cdots, T\}$, local batch size $B_{\mathrm{loc}}$

4 $t \leftarrow 0$ ;             // initialize the global iteration number
5 **for** $s = 0, \ldots, R - 1$ **do**
6     $H^{(s)} \leftarrow$ `GetH(s)` ;      // get synchronization period for the current round
7     **for** *each worker* $k$ **do in parallel**
8        $\boldsymbol{\theta}_{k,0}^{(s)} \leftarrow \bar{\boldsymbol{\theta}}^{(0)}$ ;      // maintain a local copy of the global model parameter
9        **for** $h = 0, \ldots, H^{(s)} - 1$ **do**
10           $(\xi_{k,h,1}^{(s)}, \ldots, \xi_{k,h,B_{\mathrm{loc}}}^{(s)}) \leftarrow$ `Sample()` ;      // sample a local batch
11           $\boldsymbol{g}_{k,h}^{(s)} \leftarrow \frac{1}{B_{\mathrm{loc}}} \sum_{i=1}^{B_{\mathrm{loc}}} \nabla \ell(\boldsymbol{\theta}_{k,h}^{(s)}; \xi_{k,h,i}^{(s)})$ ;      // computing the local gradient
12           $\boldsymbol{\theta}_{k,h+1}^{(s)} \leftarrow \mathrm{OPT}(\boldsymbol{\theta}_{k,h}^{(s)}, \eta_{t+h}, \boldsymbol{g}_{k,h}^{(s)})$ ;      // update the local model with optimizer OPT
13        **end**
14     **end**
15     $\bar{\boldsymbol{\theta}}^{(s+1)} \leftarrow \frac{1}{K} \sum_{k=1}^{K} \boldsymbol{\theta}_{k,H^{(s)}}^{(s)}$ ;      // All-Reduce aggregation of local model parameters
16     $t \leftarrow t + H^{(s)}$ ;      // update the global iteration number
17 **end**

---

**Sampling local batches.**  In Algorithms 1 and 2, `Sample()` returns a local batch for each worker. In our experiments, local batches are sampled without replacement at each epoch, which is standard for distributed training (Goyal et al., 2017; Lin et al., 2020; Ortiz et al., 2021). More specifically, at the beginning of each epoch, all the workers use the same random seed to draw a shared random permutation of train data points, and partition the data points evenly among the $K$ workers. Then at each local step of each worker, `Sample()` sequentially takes samples from its own partition. Once there are too few remaining samples to form a complete batch, a new permutation is sampled and a new epoch starts. For our theoretical analysis, following Gu et al. (2023), we assume `Sample()` takes samples with replacement, i.e., the $K$ workers are taking i.i.d. samples from the globally shared dataset/distribution. See Appendix B in Gu et al. (2023) for pseudocodes of sampling with and without replacement.

**Setting synchronization periods.** In Algorithm 2, GetH($s$) is a function that returns the synchronization period $H^{(s)}$ for the current round. Conventionally, $H^{(s)}$ is chosen as a fixed value, so GetH($s$) always returns a constant. In this paper, we study how $H^{(s)}$ should change as training goes on, e.g., in QSR, GetH($s$) works as specified in Section 2.

## C  EXPERIMENTAL DETAILS

This section lists the additional experimental details omitted in the main text.

**Software and platform.** We use Pytorch Distributed with NCCL backend to support multinode distributed training and use FFCV (Leclerc et al., 2022) to accelerate data loading of ImageNet.

**Sampling scheme.** We employ the "sampling without replacement" scheme, as described in Appendix B.

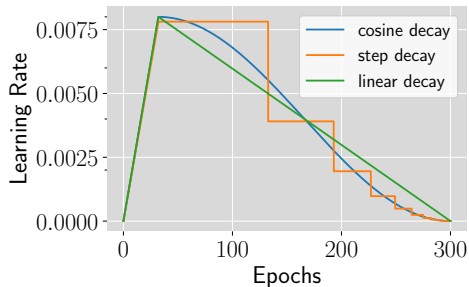

Figure 4: A visualization of the learning rate schedules we investigate.

### C.1  TRAINING DETAILS FOR RESNET-152

We generally follow the recipe in Foret et al. (2021b) to train ResNet-152. Specifically, we set the momentum as 0.9 and the weight decay $\lambda$ as 0.0001. For data augmentation, we employ random resized crop and random horizontal flip. We additionally use label smoothing 0.1. We adopt a local batch size $B_{\text{loc}} = 256$ through 8 gradient accumulations. Therefore, the batch size for BatchNorm is 32. This choice stems from our observation that a smaller batch size for BatchNorm enhances the test accuracy of parallel SGD. Since the BatchNorm statistics on each worker are estimated on the local model parameter, we pass 100 batches, each of size 32, to estimate the BatchNorm statistics on the global parameter before evaluation.

**Training details for batch size 4096.** We search the optimal peak learning rate $\eta_{\text{max}}$ of the cosine learning rate schedule among $\{0.4, 0.8, 1.6\}$ for all baseline algorithms, i.e., parallel SGD and Local SGD with constant synchronization period $H = 2$ and $H = 4$. The learning rate yielding the highest final test accuracy is selected. We find that $\eta_{\text{max}} = 0.8$ is optimal for all the baseline algorithms. For QSR with $H_{\text{base}} = 2$ and $H_{\text{base}} = 4$, we directly set $\eta_{\text{max}} = 0.8$. We search $\alpha$ among $\{0.2, 0.25, 0.3\}$, and choose $\alpha = 0.2$ and $0.25$ for QSR with $H_{\text{base}} = 2$ and $4$ respectively. Regarding other communication strategies in Figure 1(a), we set the switching point at epoch 100 and employ $H = 8$ for Post-local SGD. For $H = \beta/\eta$, we search $\beta$ among $\{0.6, 0.8, 1, 1.2\}$, finally selecting $\beta = 1$.

**Training details for batch size 16384.** The hyperparameter tuning procedure for $B = 16384$ is similar to that of $B = 4096$. We search $\eta_{\text{max}}$ among $\{0.8, 1.6, 3.2\}$ for all baseline algorithms, including SGD and Local SGD with constant synchronization period $H_{\text{base}} = 2$ and $H_{\text{base}} = 4$. We find that $\eta_{\text{max}} = 3.2$ yields the highest final test accuracy for all of them. However, for QSR, we find that peak learning rate $\eta_{\text{max}} = 3.2$ is excessively large, causing the dynamic scheduling to be triggered too late in the training process. This late triggering leaves insufficient training time for the training to fully leverage the generalization benefits introduced by local steps. Consequently, we set $\eta_{\text{max}} = 1.6$ for QSR with $H_{\text{base}} = 2$ and $4$. We search $\alpha$ among $\{0.2, 0.25, 0.3\}$, and choose $\alpha = 0.2$ for both QSR with $H_{\text{base}} = 2$ and $4$.

**Training details for the step decay scheduler.** In our experiments with step decay, we employ a batch size of 4096. Given that our step decay scheduler is derived from the cosine decay, we only need to specify the weight decay $\lambda$, and peak learning rate $\eta_{\text{max}}$. These are set identically to the values used in our cosine decay experiments. For QSR, we search the growth coefficient $\alpha$ among $\{0.2, 0.3\}$ and choose $0.2$ for both $H_{\text{base}} = 2$ and $4$.

**Training details for experiments in Appendix H.** For Local SGD + SWAP experiments in Appendix H, we use the cosine learning rate schedule with peak learning rate $\eta_{\text{max}} = 0.8$. We start with Local SGD with a constant synchronization period $H = 4$ and explore the switching point $t_0$ from $\{175, 180, 185, 190\}$.

## C.2   TRAINING DETAILS FOR VIT-B

For training ViT-B, we primarily follow the 300-epoch recipe proposed by Beyer et al. (2022). Specifically, we replace the [cls] token of the original ViT token with global average pooling and use fixed 2D sin-cos position rather than learned positional embeddings. Our implementation of the model architecture follows the high-starred repository [2] by Phil Wang. Apart from random resized crop and random horizontal flip, we employ RandAugment with parameters (2, 10) and MixUp with a coefficient of 0.2 for data augmentation. Different from Beyer et al. (2022), we use a larger batch size ($B = 4096$ or $16384$ as opposed to their $1024$) and use AdamW instead of Adam.

As for gradient clipping, we set it as 1 for standard AdamW following Beyer et al. (2022); Dosovitskiy et al. (2021) and Chen et al. (2021). However, for Local AdamW, the smaller batch size locally leads to larger gradient noise and, hence larger gradient norm for local updates. This calls for an increase in the gradient clipping threshold. We find that the

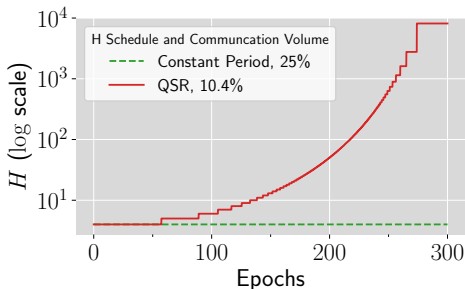

Figure 5: A visualization of the $H$ schedule for Local AdamW with a constant synchronization period $H = 4$ and with QSR $H_{\text{base}} = 4, \alpha = 0.0175$. The corresponding learning rate schedule is cosine decay with a peak learning rate of $0.008$. Adopting QSR improves the top-1 validation accuracy of Local AdamW on ViT-B from $79.32\%$ to $80.98\%$..

training process remains stable even when we remove gradient clipping (equivalent to setting the clipping threshold to $+\infty$) for most of the hyperparameter configurations we tested. For ease of tuning, we choose to turn off gradient clipping for Local AdamW unless otherwise stated.

**Training details for batch size 4096.**   We use 10k iterations for learning rate warmup following (Beyer et al., 2022; Dosovitskiy et al., 2021; Chen et al., 2021). For parallel AdamW and Local AdamW ($H = 4$), we explore combinations of $\eta_{\max}$ and weight decay $\lambda$ from the grid $\{0.05, 0.1\} \times \{0.004, 0.008, 0.016\}$. To optimize the final test accuracy, we select $\eta_{\max} = 0.008, \lambda = 0.1$ for parallel AdamW and $\eta_{\max} = 0.008, \lambda = 0.05$ for Local AdamW ($H = 4$). For Local AdamW ($H = 8$), keeping $\lambda = 0.05$, we conduct a grid search for $\eta_{\max}$ among $\{0.004, 0.008, 0.016\}$ and choose $\eta_{\max} = 0.008$. For QSR with $H_{\text{base}} = 4$ and $8$, we directly use $\eta_{\max} = 0.008$ and $\lambda = 0.05$. To optimize $\alpha$, we search among $\{0.015, 0.0175, 0.02\}$ and find $\alpha = 0.0175$ works best for both QSR with $H_{\text{base}} = 4$ and $8$. Regarding the communication strategy of $H = \beta/\eta$ in Figure 1(b), we explore $\beta$ among $\{0.025, 0.03, 0.035, 0.04\}$, settling on $\beta = 0.03$. In Figure 5, we also visualize the $H$ schedule for Local AdamW with a constant synchronization period and with QSR.

**Training details for batch size 16384.**   To keep the same portion of the total budget for learning rate warmup as $B = 4096$, we set the warmup iterations to 2.5k. We set $\lambda$ as 0.1 and 0.05 for parallel AdamW and Local AdamW, respectively. We search for the optimal $\eta_{\max}$ among $\{0.004, 0.008, 0.016\}$ and select $\eta_{\max} = 0.004$ for parallel AdamW, $\eta_{\max} = 0.016$ for Local AdamW with $H = 4$ and $8$. We adopt the same $\lambda$ and $\eta_{\max}$ as Local AdamW for QSR. For QSR with $H_{\text{base}} = 4$, we search for the optimal $\alpha$ among $\{0.015, 0.0175, 0.02\}$ and choose $\alpha = 0.0175$. For QSR with $H_{\text{base}} = 8$, we search for the optimal $\alpha$ among $\{0.01, 0.0175\}$, finally picking $\alpha = 0.01$.

**Training details for linear and step decay schedulers.**   For both step and linear decay schedulers, we employ a batch size of 4096. For the step decay scheduler, the peak learning rate $\eta_{\max}$ and weight decay $\lambda$ are set identically to the values used in our cosine decay experiments. We search the growth coefficient $\alpha$ for QSR among $\{0.015, 0.0175\}$ and choose 0.015 for both $H_{\text{base}} = 4$ and $8$. For linear decay, we use the same weight decay as our cosine decay experiments. We explore $\eta_{\max}$ values from $\{0.004, 0.008, 0.016\}$ for baselines, finally picking $\eta_{\max} = 0.008$ for parallel AdamW and $\eta_{\max} = 0.016$ for Local AdamW. For QSR, we adopt the same $\eta_{\max}$ and $\alpha$ as in our cosine decay experiments. Additionally, we add a gradient clipping threshold of $4$ for Local AdamW with a constant synchronization period to stabilize training.

---

[2]https://github.com/lucidrains/vit-pytorch

**Training details for experiments in Appendix G.** For the experiments in Table 6, we employ the same weight decay $\lambda$ and peak learning rate $\eta_{\max}$ as used in the cosine schedule. Specifically, we set $\lambda = 0.1, \eta_{\max} = 0.008$ for parallel AdamW and $\lambda = 0.05, \eta_{\max} = 0.008$ for Local AdamW. In Table 6(a), for the cubic rule, we search $\rho$ among $\{0.0025, 0.005, 0075, 0.01\}$ and opt for $\rho = 0.0075$, which gives the highest test accuracy. For QSR, we adopt the same $\alpha$ value, 0.0175, as in our cosine decay experiments. In Table 6(b), we set $\rho = 0.0075$ and $\alpha = 0.0175$ for the cubic rule and QSR, respectively, which are optimal for the original cosine decay schedule, as indicated by Figure 6. As mentioned in Section 2, the final synchronization period may be truncated. Specifically, workers are forced to synchronize at the last iteration if the last synchronization period exceeds the remaining iterations. However, the modified cosine schedule experiments seek to validate that the cubic rule can produce an overly large $H$ when the learning rate is constant. To prevent the truncation from distorting the results, we present the test accuracy at the conclusion of the last full synchronization period, which is not truncated, for both scalings.

**Training details for experiments in Appendix H.** For Local AdamW + SWAP experiments in Figure 9(b), we use the cosine learning rate schedule with peak learning rate $\eta_{\max} = 0.008$ and weight decay $\lambda = 0.05$. We start with Local AdamW with a constant synchronization period $H = 4$ and explore the switching point $t_0$ from $\{220, 240, 260, 280\}$.

# D  ADDITIONAL EXPERIMENTS ON RESNET-50

Our paper primarily focuses on training relatively large models with long horizons and proposes QSR to effectively improve the generalization while saving communication. However, on the flip side, QSR may not always yield noticeable generalization benefits for smaller models trained with shorter horizons. As shown in Table 5, for the 90-epoch training of ResNet-50 with cosine learning rate decay, the generalization benefit of QSR over Local SGD with a constant communication period is negligible. Nonetheless, training in this regime is not costly, either, making it less of a critical concern. Specifically, completing this 90-epoch training of ResNet-50 requires only 6.6 hours on a single machine equipped with 8 NVIDIA GeForce RTX 3090 GPUs. In comparison, the 300-epoch training of ViT investigated in the main text necessitates over 50 hours on the same setup.

Table 5: QSR does not yield noticeable improvement in test accuracy for the 90-epoch training of ResNet-50.

| Method | Val. Acc. (%) |
|---|---|
| Parallel SGD | 76.84 |
| Local SGD ($H = 2$) | 76.60 |
| +QSR ($H_{\text{base}} = 2$) | 76.65 |

# E    SUPPLEMENTARY MATERIALS FOR SECTION 3

## E.1    MISSING DEFINITIONS AND ASSUMPTIONS

For a function $F : \mathbb{R}^d \to \mathbb{R}^d$, we use $\partial F(\boldsymbol{\theta})$ to denote its Jacobian at $\boldsymbol{\theta}$ and use $\partial^2 F(\boldsymbol{\theta})$ to denote the second order derivative at $\boldsymbol{\theta}$. For any matrix $\boldsymbol{M} \in \mathbb{R}^{d \times d}$, $\partial^2 F(\boldsymbol{\theta})[\boldsymbol{M}] = \sum_{i \in [d]} \langle \frac{\partial^2 F_i}{\partial \boldsymbol{\theta}^2}, \boldsymbol{M} \rangle \boldsymbol{e}_i$ where $\boldsymbol{e}_i$ is the $i$-th vector of the standard basis. For convenience, we write $\partial^2 (\nabla \mathcal{L})(\boldsymbol{\theta})[\boldsymbol{M}]$ as $\nabla^3 \mathcal{L}(\boldsymbol{\theta})[\boldsymbol{M}]$.

**Assumption E.1.** *Following Gu et al. (2023), we assume that $\mathcal{L}(\boldsymbol{\theta})$ and $\boldsymbol{\Sigma}(\boldsymbol{\theta})^{1/2}$ are $\mathcal{C}^\infty$-smooth on $\mathbb{R}^d$. We also assume that $\|\nabla \ell(\boldsymbol{\theta}; \xi)\|_2$ is uniformly bounded for all $\boldsymbol{\theta}$ and $\xi$.*

**Assumption E.2.** *$\Gamma$ is a $\mathcal{C}^\infty$-smooth, $(d - m)$-dimensional compact submanifold of $\mathbb{R}^d$ such that any $\boldsymbol{\zeta} \in \Gamma$ is a local minimizer of $\mathcal{L}$ and $\mathrm{rank}(\nabla^2 \mathcal{L}(\boldsymbol{\zeta})) = m$. Additionally, there exists an open neighborhood $U$ of $\Gamma$ such that $\Gamma = \arg\min_{\boldsymbol{\theta} \in U} \mathcal{L}(\boldsymbol{\theta})$.*

Assumption E.2 is motivated by recent empirical observations that low-loss solutions on the loss landscape are not isolated but path-connected (Garipov et al., 2018; Draxler et al., 2018; Frankle et al., 2020). It is also adopted by Li et al. (2021c); Lyu et al. (2022); Gu et al. (2023).

**Definition E.1** (Gradient Flow Projection). *Fix $\boldsymbol{\theta}_{\mathrm{null}} \notin \Gamma$. For $\boldsymbol{x} \in \mathbb{R}^d$, the gradient flow starting from $\boldsymbol{x}$ is the solution to $\frac{\mathrm{d}\boldsymbol{x}(t)}{\mathrm{d}t} = -\nabla \mathcal{L}(\boldsymbol{x}(t))$ with the initial condition $\boldsymbol{x}(0) = \boldsymbol{x}$. The gradient flow projection of $\boldsymbol{x}$ is defined as $\Phi(\boldsymbol{x}) := \lim_{t \to +\infty} \boldsymbol{x}(t)$ if the limit exists and belongs to $\Gamma$. Otherwise, $\Phi(\boldsymbol{x}) := \boldsymbol{\theta}_{\mathrm{null}}$.*

**Definition E.2** (Slow SDE for SGD, formal). *Given $\boldsymbol{\zeta}_0 \in \Gamma$, define $\boldsymbol{\zeta}(t)$ as the solution to the following SDE with initial condition $\boldsymbol{\zeta}(0) = \boldsymbol{\zeta}_0$:*

$$\mathrm{d}\boldsymbol{\zeta}(t) = P_{\boldsymbol{\zeta}} \Big( \underbrace{\tfrac{1}{\sqrt{B}} \boldsymbol{\Sigma}_\|^{1/2}(\boldsymbol{\zeta}) \mathrm{d}\boldsymbol{W}_t}_{\textit{(a) diffusion on } \Gamma} \underbrace{- \tfrac{1}{2B} \nabla^3 \mathcal{L}(\boldsymbol{\zeta})[\widehat{\boldsymbol{\Sigma}}_\Diamond(\boldsymbol{\zeta})] \mathrm{d}t}_{\textit{(b) drift on } \Gamma} \Big). \tag{6}$$

*Here, for any $\boldsymbol{\zeta} \in \Gamma$, $P_{\boldsymbol{\zeta}}$ is a projection operator that maps any differential form $\boldsymbol{A}\mathrm{d}\boldsymbol{W}_t + \boldsymbol{b}\mathrm{d}t$ in Itô calculus to $\partial \Phi(\boldsymbol{\zeta})\boldsymbol{A}\mathrm{d}\boldsymbol{W}_t + \big(\partial \Phi(\boldsymbol{\zeta})\boldsymbol{b} + \frac{1}{2}\partial^2 \Phi(\boldsymbol{\zeta})[\boldsymbol{A}\boldsymbol{A}^\top]\big)$, which guarantees $\boldsymbol{\zeta}$ to remain on the manifold after taking such an infinitesimal step. $B$ is the total batch size. $\boldsymbol{\Sigma}_\|(\boldsymbol{\zeta}) := \partial \Phi(\boldsymbol{\zeta})\boldsymbol{\Sigma}(\boldsymbol{\zeta})\partial \Phi(\boldsymbol{\zeta})$ is the covariance matrix of gradient noise projected onto the tangent space of $\boldsymbol{\zeta}$ at $\Gamma$, and $\widehat{\boldsymbol{\Sigma}}_\Diamond(\boldsymbol{\zeta})$ is the noise covariance in the rest, with coordinates rescaled in the eigenbasis $\{(\lambda_i, \boldsymbol{v}_i)\}_{i=1}^d$ of $\nabla^2 \mathcal{L}(\boldsymbol{\zeta})$:*

$$\widehat{\boldsymbol{\Sigma}}_\Diamond(\boldsymbol{\zeta}) := \sum_{i,j : (\lambda_i \neq 0) \vee (\lambda_j \neq 0)} \tfrac{1}{\lambda_i + \lambda_j} \langle \boldsymbol{\Sigma}(\boldsymbol{\zeta}) - \boldsymbol{\Sigma}_\|(\boldsymbol{\zeta}), \boldsymbol{v}_i \boldsymbol{v}_j^\top \rangle \boldsymbol{v}_i \boldsymbol{v}_j^\top.$$

**Definition E.3** (Slow SDE for Local SGD with $H \sim \eta^{-1}$, formal). *Consider the scaling $H = \beta/\eta$ for some constant $\beta$. Given $\boldsymbol{\zeta}_0 \in \Gamma$, define $\boldsymbol{\zeta}(t)$ as the solution to the following SDE with initial condition $\boldsymbol{\zeta}(0) = \boldsymbol{\zeta}_0$:*

$$\mathrm{d}\boldsymbol{\zeta}(t) = P_{\boldsymbol{\zeta}} \Big( \underbrace{\tfrac{1}{\sqrt{B}} \boldsymbol{\Sigma}_\|^{1/2}(\boldsymbol{\zeta}) \mathrm{d}\boldsymbol{W}_t}_{\textit{(a) diffusion on } \Gamma} \underbrace{- \tfrac{1}{2B} \nabla^3 \mathcal{L}(\boldsymbol{\zeta})[\widehat{\boldsymbol{\Sigma}}_\Diamond(\boldsymbol{\zeta})] \mathrm{d}t}_{\textit{(b) drift on } \Gamma, \textit{ same as SGD}} \underbrace{- \tfrac{K-1}{2B} \nabla^3 \mathcal{L}(\boldsymbol{\zeta})[\widehat{\boldsymbol{\Psi}}(\boldsymbol{\zeta}; H\eta)] \mathrm{d}t}_{\textit{(c) an extra drift term on } \Gamma} \Big), \tag{7}$$

*where $K$ is the number of workers, $B, \boldsymbol{\Sigma}_\|(\boldsymbol{\zeta})$ and $\widehat{\boldsymbol{\Sigma}}_\Diamond(\boldsymbol{\zeta})$ are the same as in Definition 3.1. Here, $\widehat{\boldsymbol{\Psi}}(\boldsymbol{\zeta}; \beta)$ is a PSD matrix depending on gradient noise and Hessian defined as follows:*

$$\widehat{\boldsymbol{\Psi}}(\boldsymbol{\zeta}) := \sum_{i,j : (\lambda_i \neq 0) \vee (\lambda_j \neq 0)} \tfrac{\psi(\eta H \cdot (\lambda_i + \lambda_j))}{\lambda_i + \lambda_j} \langle \boldsymbol{\Sigma}_\Diamond(\boldsymbol{\zeta}), \boldsymbol{v}_i \boldsymbol{v}_j^\top \rangle \boldsymbol{v}_i \boldsymbol{v}_j^\top, \tag{8}$$

*where $\{\boldsymbol{v}_i\}_{i=1}^d$ is a set of eigenvectors of $\nabla^2 \mathcal{L}(\boldsymbol{\zeta})$ that forms an orthonormal eigenbasis, and $\lambda_1, \ldots, \lambda_d$ are the corresponding eigenvalues. Additionally, $\psi(x) := \frac{e^{-x} - 1 + x}{x}$ for $x \neq 0$ and $\psi(0) = 0$.*

Notice that $\psi(x)$ monotonically increases in $x$ and has the limit $\lim_{x \to 0} \psi(x) = 0$ and $\lim_{x \to \infty} \psi(x) = 1$. Therefore, given $\boldsymbol{\zeta}$, $\widehat{\boldsymbol{\Psi}}(\boldsymbol{\zeta}; \beta)$ is a monotonically increasing function of $\beta$ in the eigenspace of the Hessian matrix $\nabla^2 \mathcal{L}(\boldsymbol{\zeta})$.

### E.2 Proof for Theorem 3.1

We consider the asymptotics that $\eta \to 0, \alpha \to 0$ and $\alpha = \Omega(\eta^\gamma)$ for all $\gamma > 0$. We use big-$\mathcal{O}$ notation to hide constants independent of $\eta, \alpha$, and use big-$\tilde{\mathcal{O}}$ notations to hides constants independent of $\eta, \alpha$ and also polylog factors of $\eta, \alpha$. We define $\phi^{(s)} := \Phi(\bar{\boldsymbol{\theta}}^{(s)})$ and let $R_{\text{tot}} := \lfloor \frac{T}{H\eta^2} \rfloor = \lfloor \frac{T}{\alpha^2} \rfloor$ be the total number of rounds.

**Proof outline.** The general framework of our proof follows (Li et al., 2019a) which demonstrates the close tracking between SGD iterates and the conventional SDE by examining the moments of parameter changes over a small observation interval $\eta$. However, their analysis is not directly applicable to our case. Their SDE approximation is only valid for $\mathcal{O}(\eta^{-1})$ steps while our QSR involves multiple communication rounds, each containing $\mathcal{O}(\eta^{-2})$ steps. To tackle this challenge, we treat each round as a continuous-time observation interval of length $\alpha^2$, and then establish that the moments of changes in the manifold projection of Local SGD and the corresponding slow SDE (5), specifically the moments of $\phi^{(s+1)} - \phi^{(s)}$ and $\boldsymbol{\zeta}((s+1)\alpha^2) - \boldsymbol{\zeta}(s\alpha^2)$, are closely aligned.

Notably, though the results in (Gu et al., 2023) serve as a building block to compute the moments of $\phi^{(s+1)} - \phi^{(s)}$ in Lemmas E.1 to E.3, their analysis is not trivially extendable to QSR. This is because their analysis depends on the condition $H\eta = \mathcal{O}(1)$, and many bounds therein explode as $H\eta \to \infty$, e.g., Theorem 3.3, Lemmas I.14 and I.16 therein. In the context of QSR, where $H\eta = \frac{\alpha^2}{\eta}$ goes to infinity as $\eta$ approaches 0, the condition $H\eta = \mathcal{O}(1)$ is violated, rendering the analysis in Gu et al. (2023) ineffective for QSR.

In the following lemma, we present equivalent forms of (3), (4) and (5) that are less intuitive but more friendly to mathematical analysis.

**Theorem E.1.** *Equations (3), (4), (5) can be rewritten as the following SDEs, respectively:*

$$\mathrm{d}\boldsymbol{\zeta}(t) = \frac{1}{\sqrt{B}}\partial\Phi(\boldsymbol{\zeta})\boldsymbol{\Sigma}(\boldsymbol{\zeta})^{1/2}\mathrm{d}\boldsymbol{W}_t + \frac{1}{2B}\partial^2\Phi(\boldsymbol{\zeta})[\boldsymbol{\Sigma}(\boldsymbol{\zeta})]\mathrm{d}t, \tag{9}$$

$$\mathrm{d}\boldsymbol{\zeta}(t) = \frac{1}{\sqrt{B}}\partial\Phi(\boldsymbol{\zeta})\boldsymbol{\Sigma}(\boldsymbol{\zeta})^{1/2}\mathrm{d}\boldsymbol{W}_t + \frac{1}{2B}\partial^2\Phi(\boldsymbol{\zeta})[\boldsymbol{\Sigma}(\boldsymbol{\zeta}) + (K-1)\boldsymbol{\Psi}(\boldsymbol{\zeta})]\mathrm{d}t, \tag{10}$$

$$\mathrm{d}\boldsymbol{\zeta}(t) = \frac{1}{\sqrt{B}}\partial\Phi(\boldsymbol{\zeta})\boldsymbol{\Sigma}(\boldsymbol{\zeta})^{1/2}\mathrm{d}\boldsymbol{W}_t + \frac{K}{2B}\partial^2\Phi(\boldsymbol{\zeta})[\boldsymbol{\Sigma}(\boldsymbol{\zeta})]\mathrm{d}t. \tag{11}$$

*Proof.* Directly apply Lemmas I.1 to I.5 of Gu et al. (2023), and we have this theorem. □

Based on Gu et al. (2023)'s analysis, below we compute the moments of $\phi^{(s+1)} - \phi^{(s)}$ through a series of lemmas. Then, we follow Gu et al. (2023)'s method of moments to derive the SDE approximation.

**Lemma E.1.** *For any round $s \le R_{\text{tot}}$ and any worker $k \in [K]$, if $\phi^{(s)} \in \Gamma$, then it holds with probability at least $1 - \delta$, where $\delta = \mathcal{O}(\text{poly}(\eta))$, that $\Phi(\boldsymbol{\theta}_{k,H}^{(s)}) \in \Gamma$ and $\|\boldsymbol{\theta}_{k,H}^{(s)} - \Phi(\boldsymbol{\theta}_{k,H}^{(s)})\|_2 = \mathcal{O}(\sqrt{\eta \log \frac{1}{\eta\delta}})$.*

*Proof.* The key insight is that the dynamics of each worker before averaging in each round is just the standard SGD with a smaller batch size, $B_{\text{loc}}$. Since the distance bound to $\Gamma$, Theorem 3.3 in Gu et al. (2023), also applies to SGD by taking $K' = 1$ and $H' = \frac{1}{\eta}$, we can apply this result to obtain that $\|\boldsymbol{\theta}_{k,H}^{(s)} - \Phi(\boldsymbol{\theta}_{k,H}^{(s)})\|_2 = \mathcal{O}(\sqrt{\eta \log \frac{1}{\eta\delta}})$. □

Before computing the moments of the change in manifold projection for each worker $\Phi(\boldsymbol{\theta}_{k,H}^{(s)}) - \phi^{(s)}$, we introduce Preliminary Lemmas E.1 and E.2. Specifically, the Itô-Taylor expansion lemma E.1 is a straightforward application of Lemma B.7 of (Malladi et al., 2022) on a bounded set. Preliminary Lemma E.2 is adapted from Lemma 26 of (Li et al., 2019a).

Let $\boldsymbol{X}(t)$ be the solution to the SDE $\mathrm{d}\boldsymbol{X}(t) = \boldsymbol{b}(\boldsymbol{X}(t))\mathrm{d}t + \boldsymbol{\sigma}(\boldsymbol{X}(t))\mathrm{d}\boldsymbol{W}_t$, where $\boldsymbol{b}(\cdot) : \mathbb{R}^d \to \mathbb{R}^d$ is the drift function and $\boldsymbol{\sigma}(\cdot) : \mathbb{R}^d \to \mathbb{R}^{d \times d}$ is the diffusion matrix. Both $\boldsymbol{b}(\cdot)$ and $\boldsymbol{\sigma}(\cdot)$ belong to $\mathcal{C}^4$. Let $\mathcal{S}$ be a bounded invariant set of the SDE. That is, if $\boldsymbol{X}(0) \in \mathcal{S}$, for any $t \geq 0$, $\boldsymbol{X}(t) \in \mathcal{S}$ almost surely. Let $\eta_e$ be the "effective learning rate", which can be viewed as the length of the continuous-time observation interval for $\boldsymbol{X}(t)$. Then we have the following lemma.

**Preliminary Lemma E.1** (Itô-Taylor expansion). *Let $g : \mathbb{R}^d \to \mathbb{R}$ be any $\mathcal{C}^4$-smooth function. Define*

$$\mathcal{A}g(\boldsymbol{x}) := \sum_{i \in [D]} b_i(\boldsymbol{x})\partial_i g(\boldsymbol{x}) + \frac{1}{2} \sum_{i,j \in [D]} \left( \sum_{l \in [D]} \sigma_{i,l}(\boldsymbol{x})\sigma_{l,j}(\boldsymbol{x}) \right) \partial_{i,j}^2 g(\boldsymbol{x}). \tag{12}$$

*Given $\boldsymbol{X}(t) = \boldsymbol{x} \in \mathcal{S}$, there exists a constant $C$ independent of $\eta_e$ such that*

$$|\mathbb{E}[g(\boldsymbol{X}(t + \eta_e)) - g(\boldsymbol{x}) - \eta_e \mathcal{A}g(\boldsymbol{x})]| \leq C\eta_e^2$$

*Proof.* WLOG, we prove the case for $t = 0$. Due to the Markovian property of Itô processes, the same proof can be done for any $t > 0$ by a time shift. Give $\boldsymbol{X}(0) = \boldsymbol{x} \in \mathcal{S}$, by Itô's lemma,

$$g(\boldsymbol{X}(\eta_e)) = g(\boldsymbol{x}) + \int_0^{\eta_e} \mathcal{A}g(\boldsymbol{X}(s))\mathrm{d}s + \int_0^{\eta_e} \langle \Lambda g(\boldsymbol{X}(s)), \mathrm{d}\boldsymbol{W}_s \rangle,$$

where $\Lambda(\boldsymbol{x}) := \boldsymbol{\sigma}(\boldsymbol{x})^\top \nabla g(\boldsymbol{x})$. $\qquad \square$

Further apply Itô's lemma to $\mathcal{A}g(\boldsymbol{X}(s))$ and we have

$$g(\boldsymbol{X}(\eta_e)) = g(\boldsymbol{x}) + \int_0^{\eta_e} \left( \mathcal{A}g(\boldsymbol{x}) + \int_0^s \mathcal{A}^2 g(\boldsymbol{X}(r))\mathrm{d}r + \int_0^s \langle \Lambda\mathcal{A}g(\boldsymbol{X}(r)), \mathrm{d}\boldsymbol{W}_r \rangle \right) \mathrm{d}s$$

$$+ \int_0^{\eta_e} \langle \Lambda g(\boldsymbol{X}(s)), \mathrm{d}\boldsymbol{W}_s \rangle$$

$$= g(\boldsymbol{x}) + \eta_e \mathcal{A}g(\boldsymbol{x}) + \int_0^{\eta_e} \int_0^s \mathcal{A}^2 g(\boldsymbol{X}(r))\mathrm{d}r\mathrm{d}s$$

$$+ \int_0^{\eta_e} \int_0^s \langle \Lambda\mathcal{A}g(\boldsymbol{X}(r)), \mathrm{d}\boldsymbol{W}_r \rangle \mathrm{d}s + \int_0^{\eta_e} \langle \Lambda g(\boldsymbol{X}(s)), \mathrm{d}\boldsymbol{W}_s \rangle.$$

Take expectation on both sides, and the last two terms become zero:

$$\mathbb{E}g(\boldsymbol{X}(\eta_e)) = g(\boldsymbol{x}) + \eta_e \mathcal{A}g(\boldsymbol{x}) + \int_0^{\eta_e} \int_0^s \mathcal{A}^2 g(\boldsymbol{X}(r))\mathrm{d}r\mathrm{d}s.$$

Since $\boldsymbol{X}(s)$ belongs to the bounded set $\mathcal{S}$, there exists a constant $C$ independent of $\eta_e$ such that $|\mathcal{A}^2 g(\boldsymbol{y})| \leq C$ for all $\boldsymbol{y} \in \mathcal{S}$. Therefore,

$$|\mathbb{E}[g(\boldsymbol{X}(\eta_e)) - g(\boldsymbol{x}) - \eta_e \mathcal{A}g(\boldsymbol{x})]| \leq C\eta_e^2.$$

**Preliminary Lemma E.2** (Adaptation of Lemma 26 in (Li et al., 2019a)). *Given $\boldsymbol{X}(t) = \boldsymbol{x} \in \mathcal{S}$, denote the change in $\boldsymbol{X}(s)$ over time interval $\eta_e$ as $\tilde{\boldsymbol{\Delta}}(\boldsymbol{x}, t, \eta_e) := \boldsymbol{X}(t + \eta_e) - \boldsymbol{x}$. Then, for all $\boldsymbol{x} \in \mathcal{S}$ and $t \geq 0$, there exists a constant $C'$ independent of $\eta_e$ such that*

$$\mathbb{E}[\prod_{j=1}^{n+1} \left| \tilde{\Delta}_{i_j}(\boldsymbol{x}, t, \eta_e) \right|] \leq C'\eta_e^{\frac{n+1}{2}}, \quad \forall 1 \leq i_1, \cdots, i_{n+1} \leq d,$$

*where $n \geq 1$.*

*Proof.* WLOG, we prove the case for $t = 0$. Due to the Markovian property of Itô processes, the same proof can be done for any $t > 0$ by a time shift. Denote $\tilde{\boldsymbol{\Delta}}(\boldsymbol{x}) := \tilde{\boldsymbol{\Delta}}(\boldsymbol{x}, 0, \eta_e)$ for brevity. By definition,

$$\tilde{\boldsymbol{\Delta}}(\boldsymbol{x}) = \int_0^{\eta_e} \boldsymbol{b}(\boldsymbol{X}(s))\mathrm{d}s + \int_0^{\eta_e} \boldsymbol{\sigma}(\boldsymbol{X}(s))\mathrm{d}\boldsymbol{W}_s.$$

By triangle inequality, for all $i \in [d]$,

$$\left| \tilde{\Delta}_i(\boldsymbol{x}) \right| \le \left\| \int_0^{\eta_{\mathrm{e}}} \boldsymbol{b}(\boldsymbol{X}(s)) \mathrm{d}s \right\|_2 + \left\| \int_0^{\eta_{\mathrm{e}}} \boldsymbol{\sigma}(\boldsymbol{X}(s)) \mathrm{d}\boldsymbol{W}_s \right\|_2.$$

Therefore,

$$\mathbb{E}[\prod_{j=1}^{n+1} \left| \tilde{\Delta}_{i_j}(\boldsymbol{x}) \right|] \le \left( \mathbb{E} \left\| \int_0^{\eta_{\mathrm{e}}} \boldsymbol{b}(\boldsymbol{X}(s)) \mathrm{d}s \right\|_2 + \mathbb{E} \left\| \int_0^{\eta_{\mathrm{e}}} \boldsymbol{\sigma}(\boldsymbol{X}(s)) \mathrm{d}\boldsymbol{W}_s \right\|_2 \right)^{n+1}$$

$$\le 2^n \left( \underbrace{\mathbb{E} \left\| \int_0^{\eta_{\mathrm{e}}} \boldsymbol{b}(\boldsymbol{X}(s)) \mathrm{d}s \right\|_2}_{\mathcal{T}_1} \right)^{n+1} + 2^n \left( \underbrace{\mathbb{E} \left\| \int_0^{\eta_{\mathrm{e}}} \boldsymbol{\sigma}(\boldsymbol{X}(s)) \mathrm{d}\boldsymbol{W}_s \right\|_2}_{\mathcal{T}_2} \right)^{n+1}.$$

By triangle inequality,

$$\mathcal{T}_1 \le \mathbb{E} \left[ \int_0^{\eta_{\mathrm{e}}} \| \boldsymbol{b}(\boldsymbol{X}(s) \|_2 \mathrm{d}s \right].$$

By Cauchy-Schwarz inequality and Itô's isometry,

$$\mathcal{T}_2 \le \sqrt{\mathbb{E} \left\| \int_0^{\eta_{\mathrm{e}}} \boldsymbol{\sigma}(\boldsymbol{X}(s)) \mathrm{d}\boldsymbol{W}_s \right\|_2^2} = \sqrt{\mathbb{E} \int_0^{\eta_{\mathrm{e}}} \mathrm{tr}[\boldsymbol{\sigma}(\boldsymbol{X}(s)^\top \boldsymbol{\sigma}(\boldsymbol{X}(s))] \mathrm{d}s}.$$

Since $\boldsymbol{X}(s) \in \mathcal{S}$ almost surely and $\mathcal{S}$ is a bounded set, there exists constants $C_1$ and $C_2$ such that $\mathcal{T}_1 \le C_1 \eta_{\mathrm{e}}, \mathcal{T}_2 \le C_2 \eta_{\mathrm{e}}^{0.5}$. Substituting the bounds for $\mathcal{T}_1$ and $\mathcal{T}_2$ back, we have the lemma. $\qquad \square$

**Lemma E.2.** *For any round $s \le R_{\mathrm{tot}}$ and any worker $k \in [K]$, given $\boldsymbol{\phi}^{(s)} \in \Gamma$, then*

$$\mathbb{E}[\Phi(\boldsymbol{\theta}_{k,H}^{(s)}) - \boldsymbol{\phi}^{(s)} \mid \boldsymbol{\phi}^{(s)}] = \frac{\alpha^2}{2 B_{\mathrm{loc}}} \partial^2 \Phi(\boldsymbol{\phi}^{(s)})[\boldsymbol{\Sigma}(\boldsymbol{\phi}^{(s)})] + \mathcal{O}(\alpha^4), \tag{13}$$

$$\mathbb{E} \left[ (\Phi(\boldsymbol{\theta}_{k,H}^{(s)}) - \boldsymbol{\phi}^{(s)})(\Phi(\boldsymbol{\theta}_{k,H}^{(s)}) - \boldsymbol{\phi}^{(s)})^\top \mid \boldsymbol{\phi}^{(s)} \right] = \frac{\alpha^2}{B_{\mathrm{loc}}} \boldsymbol{\Sigma}_{\|}(\boldsymbol{\phi}^{(s)}) + \mathcal{O}(\alpha^4), \tag{14}$$

$$\mathbb{E} \left[ (\Phi(\boldsymbol{\theta}_{k,H}^{(s)}) - \boldsymbol{\phi}^{(s)})^{\otimes 3} \mid \boldsymbol{\phi}^{(s)} \right] = \mathcal{O}(\alpha^4), \tag{15}$$

$$\mathbb{E} \left[ \| \Phi(\boldsymbol{\theta}_{k,H}^{(s)}) - \boldsymbol{\phi}^{(s)} \|_2^6 \mid \boldsymbol{\phi}^{(s)} \right] = \mathcal{O}(\alpha^6). \tag{16}$$

*Proof.* Again, the key insight is that the dynamics of each worker before averaging in each round is just the standard SGD with a smaller batch size, $B_{\mathrm{loc}}$. Since the SDE approximation theorem for Local SGD, Theorem 3.2 in Gu et al. (2023), also applies to SGD by taking $K' = 1$ and $H' = \frac{1}{\eta}$, we can apply this result to obtain that, for any $\mathcal{C}^4$-smooth function $g(\boldsymbol{\theta})$, it holds for $\boldsymbol{\zeta}$ defined in (9) with the initial condition $\boldsymbol{\zeta}(0) = \boldsymbol{\phi}^{(s)}$ that

$$|\mathbb{E}[g(\Phi(\boldsymbol{\theta}_{k,H}^{(s)}))] - \mathbb{E}[g(\boldsymbol{\zeta}(T'))]| = \tilde{\mathcal{O}}(\eta^{0.25}), \tag{17}$$

where $T' = \alpha^2$ is the continuous-time observation interval.

To establish a connection between the moments of $\Phi(\boldsymbol{\theta}_{k,t}^{(s)}) - \boldsymbol{\phi}^{(s)}$ and those of $\boldsymbol{\zeta}(T') - \boldsymbol{\phi}^{(s)}$, we can let the function $g(\boldsymbol{\theta})$ to take specific forms, each returning a single coordinate of $\boldsymbol{\theta} - \boldsymbol{\phi}^{(s)}$, $(\boldsymbol{\theta} - \boldsymbol{\phi}^{(s)})(\boldsymbol{\theta} - \boldsymbol{\phi}^{(s)})^\top$, $(\boldsymbol{\theta} - \boldsymbol{\phi}^{(s)})^{\otimes 3}$ and $\| \boldsymbol{\theta} - \boldsymbol{\phi}^{(s)} \|_2^6$. For example, to relate $\Phi(\boldsymbol{\theta}_{k,H}^{(s)}) - \boldsymbol{\phi}^{(s)}$ to $\boldsymbol{\zeta}(T') - \boldsymbol{\phi}^{(s)}$, let $g(\boldsymbol{\theta}) = \langle \boldsymbol{e}_1, \boldsymbol{\theta} \rangle$ where $\boldsymbol{e}_1 = (1, 0, \cdots, 0)^\top$. Substitute $g$ into (17), and we get $| \left\langle \boldsymbol{e}_1, \mathbb{E}[\Phi(\boldsymbol{\theta}_{k,H}^{(s)}) - \boldsymbol{\phi}^{(s)}] - \mathbb{E}[\Phi(\boldsymbol{\zeta}(T')) - \boldsymbol{\phi}^{(s)}] \right\rangle | = \tilde{\mathcal{O}}(\eta^{0.25})$. We can obtain the same results for all coordinates by letting $g(\boldsymbol{\theta}) = \langle \boldsymbol{e}_i, \boldsymbol{\theta} \rangle$ for all $i \in [D]$. Therefore, $|\mathbb{E}[\Phi(\boldsymbol{\theta}_{k,H}^{(s)}) - \boldsymbol{\phi}^{(s)}] - \mathbb{E}[\boldsymbol{\zeta}(T') - \boldsymbol{\phi}^{(s)}]| = \tilde{\mathcal{O}}(\eta^{0.25})$. Similarly, we can show that the LHS of (14) to (16) are only changed by $\tilde{\mathcal{O}}(\eta^{0.25}) = o(\mathrm{poly}(\alpha))$ when replacing $\Phi(\boldsymbol{\theta}_{k,H}^{(s)})$ with $\boldsymbol{\zeta}(T')$.

Then, it suffices to compute the moments for $\boldsymbol{\zeta}(T')$ and verify that they match the RHS of (13) to (16). Since $\Gamma$ is compact and invariant for the SDE (11) (Lemma I.39 in (Gu et al., 2023)), we can apply the Itô-Taylor expansion in Preliminary Lemma E.1 with $\eta_{\mathrm{e}} = \alpha^2$, $\boldsymbol{X}(t) = \boldsymbol{\zeta}(t)$, $\boldsymbol{b}(\boldsymbol{\zeta}) = \frac{K}{2B} \partial^2 \Phi(\boldsymbol{\zeta})[\boldsymbol{\Sigma}(\boldsymbol{\zeta})]$ and $\boldsymbol{\sigma}(\boldsymbol{\zeta}) = \sqrt{\frac{1}{2B}} \partial \Phi(\boldsymbol{\zeta}) \boldsymbol{\Sigma}^{1/2}(\boldsymbol{\zeta})$.

To obtain the first moment (13), let $g(\boldsymbol{\zeta}) = \langle \boldsymbol{e}_1, \boldsymbol{\zeta} - \boldsymbol{\phi}^{(s)} \rangle$ and substitute it into (12). By Preliminary Lemma E.1, we have

$$\left| \mathbb{E}[\zeta_1(T')] - b_1(\boldsymbol{\phi}^{(s)}) \right| = \mathcal{O}(T'^2) = \mathcal{O}(\alpha^4).$$

We can repeat this process for all coordinates of $\boldsymbol{\zeta}(T')$ to obtain

$$\mathbb{E}[\boldsymbol{\zeta}(T') - \boldsymbol{\phi}^{(s)} \mid \boldsymbol{\phi}^{(s)}] = \frac{\alpha^2}{2B_{\mathrm{loc}}} \partial^2 \Phi(\boldsymbol{\phi}^{(s)})[\boldsymbol{\Sigma}(\boldsymbol{\phi}^{(s)})] + \mathcal{O}(\alpha^4), \tag{18}$$

and thus (13).

For the second moment (14), define $g^{(i,j)}(\boldsymbol{\zeta}) = \langle \boldsymbol{M}_{i,j}, (\boldsymbol{\zeta} - \boldsymbol{\phi}^{(s)})(\boldsymbol{\zeta} - \boldsymbol{\phi}^{(s)})^\top \rangle$, where $M_{i',j'} = \begin{cases} 1, (i', j') = (i, j), \\ 0, \text{otherwise} \end{cases}$. Since $\partial_{i'} g^{(i,j)}(\boldsymbol{\zeta}) = 0$ for all $i'$, the first term of $\mathcal{A}g^{(i,j)}(\boldsymbol{\zeta})$ vanishes. It suffices to compute the second term. When $i = j$, $\partial^2_{i',j'} g^{(i,i)}(\boldsymbol{\zeta}) = \begin{cases} 2, (i', j') = (i, i) \\ 0, \text{otherwise} \end{cases}$. Therefore,

$$\mathcal{A}g^{(i,i)}(\boldsymbol{\zeta}) = \sum_{l \in [D]} \sigma_{i,l}(\boldsymbol{\zeta}) \sigma_{l,i}(\boldsymbol{\zeta}), \quad \forall i \in [D]. \tag{19}$$

When $i \neq j$, $\partial^2_{i',j'} g^{(i,j)}(\boldsymbol{\zeta}) = \begin{cases} 1, (i', j') \in \{(i, j), (j, i)\} \\ 0, \text{otherwise} \end{cases}$. Therefore,

$$\mathcal{A}g^{(i,j)}(\boldsymbol{\zeta}) = \sum_{l \in [D]} \sigma_{i,l}(\boldsymbol{\zeta}) \sigma_{l,j}(\boldsymbol{\zeta}), \quad i \neq j. \tag{20}$$

Combining (19) and (20) and noticing that $g^{(i,j)}(\boldsymbol{\phi}^{(s)}) = 0$ for all $i, j$, we have

$$\mathbb{E}[(\boldsymbol{\zeta}(T') - \boldsymbol{\phi}^{(s)})(\boldsymbol{\zeta}(T') - \boldsymbol{\phi}^{(s)})^\top \mid \boldsymbol{\phi}^{(s)}] = \frac{\alpha^2}{B} \boldsymbol{\Sigma}_\|(\boldsymbol{\phi}^{(s)}) + \mathcal{O}(\alpha^4), \tag{21}$$

and thus (14).

For the third moment (15), define $g^{(i,j,l)}(\boldsymbol{\zeta}) = \langle \boldsymbol{e}_i \otimes \boldsymbol{e}_j \otimes \boldsymbol{e}_l, (\Phi(\boldsymbol{\theta}^{(s)}_{k,H}) - \boldsymbol{\phi}^{(s)})^{\otimes 3} \rangle$. Noticing that $\partial_{i'} g^{(i,j,l)}(\boldsymbol{\phi}^{(s)}) = 0$ for all $i'$ and $\partial^2_{i',j'} g^{(i,j,l)}(\boldsymbol{\phi}^{(s)}) = 0$ for all $(i', j')$, we have

$$\mathbb{E}\left[ (\boldsymbol{\zeta}(T') - \boldsymbol{\phi}^{(s)})^{\otimes 3} \mid \boldsymbol{\phi}^{(s)} \right] = \mathcal{O}(\alpha^4), \tag{22}$$

and thus (15).

Finally, by directly applying Preliminary Lemma E.2, we have

$$\mathbb{E}\left[ \|\boldsymbol{\zeta}(T') - \boldsymbol{\phi}^{(s)}\|_2^6 \mid \boldsymbol{\phi}^{(s)} \right] = \mathcal{O}(\alpha^6) \tag{23}$$

and thus (16). $\qquad \square$

Now we are ready to compute the moments for $\boldsymbol{\phi}^{(s+1)} - \boldsymbol{\phi}^{(s)}$ at each round:

**Lemma E.3.** *For any round $s \leq R_{\mathrm{tot}}$, given $\boldsymbol{\phi}^{(s)} \in \Gamma$, then*

$$\mathbb{E}[\boldsymbol{\phi}^{(s+1)} - \boldsymbol{\phi}^{(s)} \mid \boldsymbol{\phi}^{(s)}] = \frac{\alpha^2}{2B_{\mathrm{loc}}} \partial^2 \Phi(\boldsymbol{\phi}^{(s)})[\boldsymbol{\Sigma}(\boldsymbol{\phi}^{(s)})] + \mathcal{O}(\alpha^4), \tag{24}$$

$$\mathbb{E}\left[ (\boldsymbol{\phi}^{(s+1)} - \boldsymbol{\phi}^{(s)})(\boldsymbol{\phi}^{(s+1)} - \boldsymbol{\phi}^{(s)})^\top \mid \boldsymbol{\phi}^{(s)} \right] = \frac{\alpha^2}{B} \boldsymbol{\Sigma}_\|(\boldsymbol{\phi}^{(s)}) + \mathcal{O}(\alpha^4), \tag{25}$$

$$\mathbb{E}\left[ \|\boldsymbol{\phi}^{(s+1)} - \boldsymbol{\phi}^{(s)}\|_2^6 \mid \boldsymbol{\phi}^{(s)} \right] = \mathcal{O}(\alpha^6). \tag{26}$$

*Proof.* Let $\boldsymbol{\Delta}_1 := \frac{1}{K}\sum_{k=1}^{K}\Phi(\boldsymbol{\theta}_{k,H}^{(s)}) - \boldsymbol{\phi}^{(s)}$. By Lemma E.2,

$$\mathbb{E}[\boldsymbol{\Delta}_1] = \frac{1}{K}\sum_{k=1}^{K}\mathbb{E}[\Phi(\boldsymbol{\theta}_{k,H}^{(s)}) - \boldsymbol{\phi}^{(s)}]$$

$$= \frac{\alpha^2}{2B_{\text{loc}}}\partial^2\Phi(\boldsymbol{\phi}^{(s)})[\boldsymbol{\Sigma}(\boldsymbol{\phi}^{(s)})] + \mathcal{O}(\alpha^4),$$

$$\mathbb{E}[\boldsymbol{\Delta}_1\boldsymbol{\Delta}_1^\top] = \frac{1}{K^2}\sum_{j=1}^{K}\sum_{k=1}^{K}\mathbb{E}\left[(\Phi(\boldsymbol{\theta}_{j,H}^{(s)}) - \boldsymbol{\phi}^{(s)})(\Phi(\boldsymbol{\theta}_{k,H}^{(s)}) - \boldsymbol{\phi}^{(s)})^\top\right]$$

$$= K\cdot\frac{\alpha^2}{B}\boldsymbol{\Sigma}_{\|}(\boldsymbol{\phi}^{(s)}) + \mathcal{O}(\alpha^4) + K(K-1)\cdot\mathcal{O}(\alpha^4)$$

$$= \frac{\alpha^2}{B_{\text{loc}}}\boldsymbol{\Sigma}_{\|}(\boldsymbol{\phi}^{(s)}) + \mathcal{O}(\alpha^4).$$

Let $\boldsymbol{\Delta}_2 := \frac{1}{K}\sum_{k=1}^{K}\boldsymbol{\theta}_{k,H}^{(s)} - \boldsymbol{\phi}^{(s)}$. Then $\boldsymbol{\Delta}_2 = \boldsymbol{\Delta}_1 + \frac{1}{K}\sum_{k=1}^{K}(\boldsymbol{\theta}_{k,H}^{(s)} - \Phi(\boldsymbol{\theta}_{k,H}^{(s)}))$. Finally, let $\boldsymbol{\Delta}_3 := \Phi(\frac{1}{K}\sum_{k=1}^{K}\boldsymbol{\theta}_{k,H}^{(s)}) - \boldsymbol{\phi}^{(s)}$. By Lemma E.1 it holds with probability at least $1-\delta$ that $\|\boldsymbol{\theta}_{k,H}^{(s)} - \Phi(\boldsymbol{\theta}_{k,H}^{(s)})\|_2 = \mathcal{O}(\sqrt{\eta\log\frac{1}{\eta}})$ and thus $\|\boldsymbol{\Delta}_2 - \boldsymbol{\Delta}_1\|_2 = \mathcal{O}(\sqrt{\eta\log\frac{1}{\eta}})$. Let $\delta = \eta^{100}$. Since $\|\partial\Phi(\cdot)\|_2$ is always bounded by $\mathcal{O}(1)$, we can always add an error of $\mathcal{O}(\delta)$ to our bounds for the moments and ignore the possibility that this event does not happen. To prove (24), we do Taylor expansion of $\Phi$ at $\boldsymbol{\phi}^{(s)}$, then

$$\mathbb{E}[\boldsymbol{\Delta}_3] = \mathbb{E}[\Phi(\boldsymbol{\phi}^{(s)} + \boldsymbol{\Delta}_2) - \boldsymbol{\phi}^{(s)}]$$

$$= \mathbb{E}[\Phi(\boldsymbol{\phi}^{(s)} + \boldsymbol{\Delta}_1) - \boldsymbol{\phi}^{(s)}] + \mathcal{O}\left(\sqrt{\eta\log\frac{1}{\eta}} + \delta\right)$$

$$= \mathbb{E}\left[\partial\Phi(\boldsymbol{\phi}^{(s)})\boldsymbol{\Delta}_1 + \mathcal{O}(\|\boldsymbol{\Delta}_1\|_2^2)\right] + \mathcal{O}\left(\sqrt{\eta\log\frac{1}{\eta}} + \delta\right)$$

$$= \frac{\alpha^2}{2B_{\text{loc}}}\partial^2\Phi(\boldsymbol{\phi}^{(s)})[\boldsymbol{\Sigma}(\boldsymbol{\phi}^{(s)})] + \mathcal{O}(\alpha^4).$$

The last equation uses the fact that $\partial\Phi(\boldsymbol{\phi})$, for $\boldsymbol{\phi}\in\Gamma$, is a projection matrix onto the tangent space of $\Gamma$ at $\boldsymbol{\theta}$ (Lemma 4.3 of (Li et al., 2021c)).

To prove (25), again we do Taylor expansion of $\Phi$ at $\boldsymbol{\phi}^{(s)}$ to connect $\Phi(\boldsymbol{\phi}^{(s)} + \boldsymbol{\Delta}_2)$ with $\Phi(\boldsymbol{\phi}^{(s)} + \boldsymbol{\Delta}_1)$ and obtain:

$$\mathbb{E}[\boldsymbol{\Delta}_3\boldsymbol{\Delta}_3^\top] = \mathbb{E}\left[(\Phi(\boldsymbol{\phi}^{(s)} + \boldsymbol{\Delta}_2) - \boldsymbol{\phi}^{(s)})(\Phi(\boldsymbol{\phi}^{(s)} + \boldsymbol{\Delta}_2) - \boldsymbol{\phi}^{(s)})^\top\right]$$

$$= \mathbb{E}\left[(\Phi(\boldsymbol{\phi}^{(s)} + \boldsymbol{\Delta}_1) - \boldsymbol{\phi}^{(s)})(\Phi(\boldsymbol{\phi}^{(s)} + \boldsymbol{\Delta}_1) - \boldsymbol{\phi}^{(s)})^\top\right] + \mathcal{O}\left(\sqrt{\eta\log\frac{1}{\eta}} + \delta\right).$$

Applying the second-order Taylor expansion gives

$$\mathbb{E}[\boldsymbol{\Delta}_3\boldsymbol{\Delta}_3^\top] = \mathbb{E}\bigg[\left(\partial\Phi(\boldsymbol{\phi}^{(s)})\boldsymbol{\Delta}_1\right)\left(\partial\Phi(\boldsymbol{\phi}^{(s)})\boldsymbol{\Delta}_1\right)^\top$$

$$+ \left(\partial^2\Phi(\boldsymbol{\phi}^{(s)})[\boldsymbol{\Delta}_1,\boldsymbol{\Delta}_1]\partial\Phi(\boldsymbol{\phi}^{(s)})\boldsymbol{\Delta}_1 + \boldsymbol{\Delta}_1^\top\partial\Phi(\boldsymbol{\phi}^{(s)})\partial^2\Phi(\boldsymbol{\phi}^{(s)})[\boldsymbol{\Delta}_1,\boldsymbol{\Delta}_1]^\top\right)$$

$$+ \mathcal{O}(\|\boldsymbol{\Delta}_1\|_2^4)\bigg] + \mathcal{O}\left(\sqrt{\eta\log\frac{1}{\eta}} + \delta\right).$$

By (15) and the fact that $\|\partial^2\Phi(\boldsymbol{\phi}^{(s)})\|_2$ is bounded, the above equation can be simplified to

$$\mathbb{E}[\boldsymbol{\Delta}_3\boldsymbol{\Delta}_3^\top] = \mathbb{E}\left[\partial\Phi(\boldsymbol{\phi}^{(s)})\boldsymbol{\Delta}_1\boldsymbol{\Delta}_1^\top\partial\Phi(\boldsymbol{\phi}^{(s)})\right] + \mathcal{O}(\alpha^4) + \mathcal{O}(\alpha^4) + \mathcal{O}\left(\sqrt{\eta\log\frac{1}{\eta}} + \delta\right)$$

$$= \frac{\alpha^2}{B}\boldsymbol{\Sigma}_{\|}(\boldsymbol{\phi}^{(s)}) + \mathcal{O}(\alpha^4).$$

Finally, for (26), we can repeat the above process to bound $\mathbb{E}[\|\boldsymbol{\Delta}_1\|_2^3]$, and then conclude that $\mathbb{E}[\|\Delta_3\|_2^3] = \mathcal{O}(\alpha^6)$. $\qquad\square$

Now we are ready to prove our main theorem.

*Proof for Theorem 3.1.* Let $\boldsymbol{\zeta}(t)$ be the solution of (11). Let $r$ be some integer greater than $s$. If $\phi^{(s)} \in \Gamma$, define $\hat{\boldsymbol{\zeta}}_{s,r}$ as the random variable sampled from the distribution of $\boldsymbol{\zeta}(\alpha^2 r)$ conditioned on $\boldsymbol{\zeta}(\alpha^2 s) = \phi^{(s)}$. If $\phi^{(s)} = \boldsymbol{\theta}_{\text{null}} \notin \Gamma$, define $\hat{\boldsymbol{\zeta}}_{s,r} = \mathbf{0}$.

If $\boldsymbol{\theta} \in \Gamma$, define $u(\boldsymbol{\theta}, t_1, t_2)$ the expected value of $g(\boldsymbol{\zeta}(t_2))$ conditioned on $\boldsymbol{\zeta}(t_1) = \boldsymbol{\theta}$. If $\boldsymbol{\theta} = \boldsymbol{\theta}_{\text{null}}$, define $u(\boldsymbol{\theta}, t_1, t_2) = \mathbf{0}$. That is,

$$u(\boldsymbol{\theta}, t_1, t_2) := \begin{cases} \mathbb{E}[g(\boldsymbol{\zeta}(t_2)) \mid \boldsymbol{\zeta}(t_1) = \boldsymbol{\theta}], & \boldsymbol{\theta} \in \Gamma, \\ \mathbf{0}, & \boldsymbol{\theta} \notin \Gamma. \end{cases}$$

For all $n \leq R_{\text{tot}}$, we have

$$|\mathbb{E}[g(\phi^{(n)})] - \mathbb{E}[g(\boldsymbol{\zeta}(n\alpha^2))]| = |\mathbb{E}[g(\hat{\boldsymbol{\zeta}}_{n,n})] - \mathbb{E}[g(\hat{\boldsymbol{\zeta}}_{0,n})]|$$

$$\leq \sum_{s=0}^{n-1} |\mathbb{E}[g(\hat{\boldsymbol{\zeta}}_{s+1,n})] - \mathbb{E}[g(\hat{\boldsymbol{\zeta}}_{s,n})]|$$

$$= \sum_{s=0}^{n-1} \left| \mathbb{E}\left[ \mathbb{E}[g(\hat{\boldsymbol{\zeta}}_{s+1,n}) \mid \phi^{(s+1)}] \right] - \underbrace{\mathbb{E}\left[ \mathbb{E}[g(\hat{\boldsymbol{\zeta}}_{s,n}) \mid \phi^{(s)}] \right]}_{\mathcal{T}_s} \right|.$$

By the law of total expectation and the Markovian property of Itô process,

$$\mathcal{T}_s = \mathbb{E}\left[ \mathbb{E}[g(\boldsymbol{\zeta}(n\alpha^2)) \mid \boldsymbol{\zeta}(s\alpha^2) = \phi^{(s)}] \right]$$

$$= \mathbb{E}\left\{ \mathbb{E}\left[ \mathbb{E}[g(\boldsymbol{\zeta}(n\alpha^2)) \mid \boldsymbol{\zeta}((s+1)\alpha^2)] \Big| \boldsymbol{\zeta}(s\alpha^2) = \phi^{(s)} \right] \right\}$$

$$= \mathbb{E}[u(\hat{\boldsymbol{\zeta}}_{s,s+1}, (s+1)\alpha^2, n\alpha^2)].$$

Therefore,

$$|\mathbb{E}[g(\phi^{(n)})] - \mathbb{E}[g(\boldsymbol{\zeta}(n\alpha^2))]| = \sum_{s=0}^{n-1} \left| \underbrace{\mathbb{E}[u(\phi^{(s+1)}, (s+1)\alpha^2, n\alpha^2)] - \mathbb{E}[u(\hat{\boldsymbol{\zeta}}_{s,s+1}, (s+1)\alpha^2, n\alpha^2)]}_{\mathcal{T}_s'} \right|.$$

By the law of total expectation,

$$|\mathcal{T}_s'| \leq \underbrace{|\mathbb{E}[u(\phi^{(s+1)}, (s+1)\alpha^2, n\alpha^2) - u(\hat{\boldsymbol{\zeta}}_{s,s+1}, (s+1)\alpha^2, n\alpha^2) \mid \phi^{(s)}, \phi^{(s+1)} \in \Gamma]|}_{\mathcal{A}_s}$$

$$+ |u(\mathbf{0})|\mathbb{P}(\phi^{(s)} \notin \Gamma \text{ or } \phi^{(s+1)} \notin \Gamma),$$

where the latter term comes from the definition of $u(\boldsymbol{\theta}, t_1, t_2)$ and $\hat{\boldsymbol{\zeta}}_{s,r}$. By Lemma I.11 in (Gu et al., 2023), there exists a constant $\epsilon$ such that if $\min_{\phi \in \Gamma} \|\boldsymbol{\theta} - \phi\|_2 \leq \epsilon$, then $\Phi(\boldsymbol{\theta}) \in \Gamma$. Therefore, substituting $\delta = \eta^{100}$ into Lemma E.1, we can conclude that the latter term is at most $\mathcal{O}(\eta^{100})$.

For $\mathcal{A}_s$, notice that the two terms differ only in the first position. By Lemma E.3 and (18) to (23), the moments of $\phi^{(s+1)} - \phi^{(s)}$ and $\hat{\boldsymbol{\zeta}}_{s,s+1} - \phi^{(s)}$ are close to each other. Therefore, it suffices to discuss the smoothness of $u$ and perform Taylor expansion. By Proposition 25 of (Li et al., 2019a), since $g \in \mathcal{C}^4$, $u(\phi, t_1, t_2)$ satisfies the compatibility condition for the Whitney Extension Theorem for $\phi \in \Gamma$. Therefore, there exists a function $\tilde{u}(\phi, t_1, t_2)$ that is $\mathcal{C}^4$ in $\phi$ for $\phi \in \mathbb{R}^d$ and satisfies $\tilde{u}(\phi, t_1, t_2) = u(\phi, t_1, t_2)$ for all $\phi \in \Gamma$. Denote $\tilde{u}(\phi, s\alpha^2, n\alpha^2)$ as $\tilde{u}_{s,n}(\phi)$ for brevity. Now, we can safely substitute $u$ in $\mathcal{A}_s$ with $\tilde{u}$ and perform Taylor expansion:

$$\mathcal{A}_s = \underbrace{\mathbb{E}[\tilde{u}(\phi^{(s)} + (\phi^{(s+1)} - \phi^{(s)})), (s+1)\alpha^2, n\alpha^2) \mid \phi^{(s)}, \phi^{(s+1)} \in \Gamma]}_{\mathcal{A}_s'}$$

$$- \underbrace{\mathbb{E}[\tilde{u}(\phi^{(s)} + (\hat{\boldsymbol{\zeta}}_{s,s+1} - \phi^{(s)})), (s+1)\alpha^2, n\alpha^2) \mid \phi^{(s)} \in \Gamma]}_{\mathcal{A}_s''}.$$

$$\mathcal{A}'_s = \tilde{u}(\boldsymbol{\phi}^{(s)}) + \left\langle \partial \tilde{u}_{s+1,n}(\boldsymbol{\phi}^{(s)}), \mathbb{E}[\boldsymbol{\phi}^{(s+1)} - \boldsymbol{\phi}^{(s)} \mid \boldsymbol{\phi}^{(s)}, \boldsymbol{\phi}^{(s+1)} \in \Gamma] \right\rangle$$
$$+ \frac{1}{2} \left\langle \partial^2 \tilde{u}_{s+1,n}(\boldsymbol{\phi}^{(s)}), \mathbb{E}[(\boldsymbol{\phi}^{(s+1)} - \boldsymbol{\phi}^{(s)})(\boldsymbol{\phi}^{(s+1)} - \boldsymbol{\phi}^{(s)})^\top \mid \boldsymbol{\phi}^{(s)}, \boldsymbol{\phi}^{(s+1)} \in \Gamma] \right\rangle$$
$$+ \mathcal{O}(\mathbb{E}[\|\boldsymbol{\phi}^{(s+1)} - \boldsymbol{\phi}^{(s)}\|_2^3 \mid \boldsymbol{\phi}^{(s)}, \boldsymbol{\phi}^{(s+1)} \in \Gamma]).$$

$$\mathcal{A}''_s = \tilde{u}(\boldsymbol{\phi}^{(s)}) + \left\langle \partial \tilde{u}_{s+1,n}(\boldsymbol{\phi}^{(s)}), \mathbb{E}[\hat{\boldsymbol{\zeta}}_{s,s+1} - \boldsymbol{\phi}^{(s)} \mid \boldsymbol{\phi}^{(s)} \in \Gamma] \right\rangle$$
$$+ \frac{1}{2} \left\langle \partial^2 \tilde{u}_{s+1,n}(\boldsymbol{\phi}^{(s)}), \mathbb{E}[(\hat{\boldsymbol{\zeta}}_{s,s+1} - \boldsymbol{\phi}^{(s)})(\hat{\boldsymbol{\zeta}}_{s,s+1} - \boldsymbol{\phi}^{(s)})^\top \mid \boldsymbol{\phi}^{(s)} \in \Gamma] \right\rangle$$
$$+ \mathcal{O}(\mathbb{E}[\|\hat{\boldsymbol{\zeta}}_{s,s+1} - \boldsymbol{\phi}^{(s)}\|_2^3 \mid \boldsymbol{\phi}^{(s)} \in \Gamma])$$

Substituting in $\delta = \eta^{100}$ into Lemma E.1, we can conclude that, given $\boldsymbol{\phi}^{(s)} \in \Gamma$, the event $\{\boldsymbol{\phi}^{(s+1)} \in \Gamma\}$ happens with probability at least $1 - \eta^{100}$. We can replace the condition $\boldsymbol{\phi}^{(s)}, \boldsymbol{\phi}^{(s+1)} \in \Gamma$ with $\boldsymbol{\phi}^{(s)} \in \Gamma$ in $\mathcal{A}'_s$ with an error of only $\mathcal{O}(\eta^{100})$. Therefore,

$$\mathcal{A}_s = \left\langle \partial \tilde{u}_{s+1,n}(\boldsymbol{\phi}^{(s)}), \mathbb{E}[(\boldsymbol{\phi}^{(s+1)} - \boldsymbol{\phi}^{(s)}) - (\hat{\boldsymbol{\zeta}}_{s,s+1} - \boldsymbol{\phi}^{(s)}) \mid \boldsymbol{\phi}^{(s)} \in \Gamma] \right\rangle$$
$$+ \frac{1}{2} \left\langle \partial^2 \tilde{u}_{s+1,n}(\boldsymbol{\phi}^{(s)}), \mathbb{E}[((\boldsymbol{\phi}^{(s)} - \boldsymbol{\phi}^{(s)})(\hat{\boldsymbol{\zeta}}_{s,s+1} - \boldsymbol{\phi}^{(s)})^\top \right.$$
$$\left. - (\hat{\boldsymbol{\zeta}}_{s,s+1} - \boldsymbol{\phi}^{(s)})(\hat{\boldsymbol{\zeta}}_{s,s+1} - \boldsymbol{\phi}^{(s)})^\top \mid \boldsymbol{\phi}^{(s)} \in \Gamma] \right\rangle$$
$$+ \mathcal{O}(\mathbb{E}[\|\boldsymbol{\phi}^{(s+1)} - \boldsymbol{\phi}^{(s)}\|_2^3 \mid \boldsymbol{\phi}^{(s)} \in \Gamma]) + \mathcal{O}(\mathbb{E}[\|\hat{\boldsymbol{\zeta}}_{s,s+1} - \boldsymbol{\phi}^{(s)}\|_2^3 \mid \boldsymbol{\phi}^{(s)} \in \Gamma]) + \mathcal{O}(\eta^{100}).$$

Since $\boldsymbol{\phi}^{(s)} \in \Gamma$ where $\Gamma$ is a compact set, both $\|\partial \tilde{u}_{s+1,n}(\boldsymbol{\phi}^{(s)})\|_2$ and $\|\partial^2 \tilde{u}_{s+1,n}(\boldsymbol{\phi}^{(s)})\|_2$ are bounded. Substituting Lemma E.3 and (18) to (23) to the expression of $\mathcal{A}_s$, we have $\mathcal{A}_s = \mathcal{O}(\alpha^4)$ and thus $|\mathcal{T}'_s| = \mathcal{O}(\alpha^4)$. Summing $|\mathcal{T}'_s|$ up, we have $|\mathbb{E}[g(\boldsymbol{\phi}^{(n)})] - \mathbb{E}[g(\boldsymbol{\zeta}(n\alpha^2))]| \leq \mathcal{O}(n\alpha^4) \leq \mathcal{O}(\alpha^2)$, which completes the proof. □

## F    DETAILS FOR COMMUNICATION TIME MEASUREMENT

It is straightforward to measure the time duration for the entire training, but it is hard to directly measure the communication time due to the asynchronous nature of CUDA computation. Hence, in our experiments, we derive the communication time from the difference in total training time across runs with various communication frequencies.

Specifically, let $T_{\text{para}}^{\text{tot}}, T_{H_1}^{\text{tot}}$ be the total time durations of data parallel approaches and local gradient methods with $H = H_1$, respectively. Also let $T_{\text{para}}^{\text{comm}}, T_{H_1}^{\text{comm}}$ be their communication times, and $T_{\text{para}}^{\text{comp}}, T_{H_1}^{\text{comp}}$ be their computation time. Ideally, setting the synchronization period to $H_1$ reduces the communication volume exactly by a factor of $\frac{1}{H_1}$, so these variables satisfy the following relationships:

$$T_{H_1}^{\text{comp}} = T_{\text{para}}^{\text{comp}},$$
$$T_{H_1}^{\text{comm}} = \frac{1}{H_1} T_{\text{para}}^{\text{comm}},$$
$$T_{H_1}^{\text{comm}} + T_{H_1}^{\text{comp}} = T_{H_1}^{\text{tot}},$$
$$T_{\text{para}}^{\text{comm}} + T_{\text{para}}^{\text{comp}} = T_{\text{para}}^{\text{tot}}.$$

Then we can express the communication and computation times in terms of the total time duration $T_{\text{para}}^{\text{tot}}$ and $T_{H_1}^{\text{tot}}$:

$$T_{\text{para}}^{\text{comm}} = H_1 T_{H_1}^{\text{comm}} = \frac{H_1}{H_1 - 1}(T_{\text{para}}^{\text{tot}} - T_{H_1}^{\text{tot}}),$$
$$T_{\text{para}}^{\text{comp}} = T_{H_1}^{\text{comp}} = T_{\text{para}}^{\text{tot}} - T_{\text{para}}^{\text{comm}} = \frac{H_1}{H_1 - 1} T_{H_1}^{\text{tot}} - \frac{1}{H_1 - 1} T_{\text{para}}^{\text{tot}}.$$

Therefore, we empirically measure the total time duration $\widetilde{T}^{\text{tot}}_{\text{para}}$ and $\widetilde{T}^{\text{tot}}_{H_1}$ for some $H_1$, then use the following formulas to obtain estimates of the communication and computation times:

$$\widetilde{T}^{\text{comm}}_{\text{para}} = \frac{H_1}{H_1 - 1}(\widetilde{T}^{\text{tot}}_{\text{para}} - \widetilde{T}^{\text{tot}}_{H_1}), \tag{27}$$

$$\widetilde{T}^{\text{comp}}_{\text{para}} = \frac{H_1}{H_1 - 1}\widetilde{T}^{\text{tot}}_{H_1} - \frac{1}{H_1 - 1}\widetilde{T}^{\text{tot}}_{\text{para}}. \tag{28}$$

These estimates are very predictive for the total time duration of local gradient methods with a different $H$. For example, when $H = H_2$, we can predict the total time duration $T^{\text{tot}}_{H_2}$ as follows:

$$T^{\text{comm}}_{H_2} \approx \frac{1}{H_2}\widetilde{T}^{\text{comm}}_{\text{para}}, \tag{29}$$

$$T^{\text{tot}}_{H_2} \approx \frac{1}{H_2}\widetilde{T}^{\text{comm}}_{\text{para}} + \widetilde{T}^{\text{comp}}_{\text{para}}. \tag{30}$$

We find that the relative error $\frac{|\widetilde{T}^{\text{tot}}_{H_2} - T^{\text{tot}}_{H_2}|}{T^{\text{tot}}_{H_2}} \times 100\%$, where $T^{\text{tot}}_{H_2}$ denotes the measured total time, is only $\sim 1\%$ across all configurations in Table 4, where we set $H_1 = 2, H_2 = 4$ for ResNet-152 and $H_1 = 4, H_2 = 8$ for ViT-B. The small relative error suggests that our method offers a close approximation to the actual time. For this reason, in Table 4, we report the communication time estimated by (27) and (29) for data-parallel approaches and local gradient methods with a constant synchronization period.

For QSR, since its communication volume relative to data parallel approaches, denoted as $f_{\text{QSR}}$, can be easily computed given the learning rate schedule, the growth coefficient $\alpha$ and the base synchronization period $H_{\text{base}}$, we can estimate its communication time $T^{\text{comm}}_{\text{QSR}}$ in a similar vein to (27) and (29):

$$T^{\text{comm}}_{\text{QSR}} \approx f_{\text{QSR}}\widetilde{T}^{\text{comm}}_{\text{para}}. \tag{31}$$

We report the communication time estimated by (31) in Table 4 for QSR.

## G  DISCUSSION ON MORE AGGRESSIVE SCALINGS

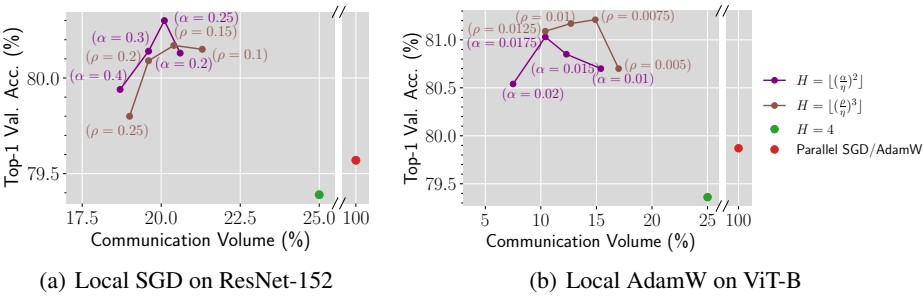

(a) Local SGD on ResNet-152    (b) Local AdamW on ViT-B

Figure 6: The cubic rule $H = \lfloor (\frac{\rho}{\eta})^3 \rfloor$ with a properly tuned $\rho$ can either outperform or underperform the QSR in test accuracy, depending on the training scenarios.

Apart from the scalings discussed in Section 3, one can consider more aggressive scalings, e.g., $H = \lfloor (\rho/\eta)^{-3} \rfloor$. Compared with QSR $H = \lfloor (\alpha/\eta)^{-2} \rfloor$ that uses the same amount of communication, this cubic synchronization rule communicates more frequently at earlier stages but much less at later stages. Our theory in Theorem 3.1 suggests that taking $H \sim \eta^{-3}$ blows up the approximation error, but as shown in Figure 6, this cubic rule with a properly tuned $\rho$ can either outperform or underperform the QSR in test accuracy, depending on the training scenarios.

We argue that this is because our quasistatic view may break in the very late phase of cosine decay, where the learning rate decays so fast that $\eta_t$ sees significant decay within a single communication round. As an example where the cubic rule performs better, we plot in Figure 8 the test accuracy

Table 6: We validate that the higher test accuracy achieved by $H \sim \eta^{-3}$ relies on the rapid decaying learning rate within a synchronization period via ablation studies on ViT-B. In Table 6(a), we replace the cosine decay schedule with a variant of the step decay schedule in Smith et al. (2020). In Table 6(b), we run both scalings with a modified cosine decay schedule that ceases to decay at some epoch $t''$. QSR consistently outperforms $H \sim \eta^{-3}$ in both cases.

(a) Local AdamW with step decay.

| Method | Val. Acc. (%) | Comm. (%) |
|---|---|---|
| Parallel AdamW | 78.51 | 100 |
| Local AdamW ($H$=4) | 78.70 | 25 |
| +QSR ($H_{\text{base}} = 4$) | **80.99** | **13.2** |
| +$H \sim \eta^{-3}$ ($H_{\text{base}} = 4$) | 80.86 | 14.4 |

(b) Local AdamW with modified cosine decay. Both scalings use $H_{\text{base}} = 4$.

| Method | $t''$ | Val. Acc. (%) |
|---|---|---|
| QSR | 260 | **80.75** |
| $H \sim \eta^{-3}$ | 260 | 80.51 |
| QSR | 250 | **80.34** |
| $H \sim \eta^{-3}$ | 250 | 79.91 |
| QSR | 240 | **79.85** |
| $H \sim \eta^{-3}$ | 240 | 79.72 |

curves of the QSR and cubic rule for training ViT-B with Local AdamW and batch size 4096. Same as our experiment setup in Section 4, the learning rate peaks at the value 0.008 and then decays to nearly zero ($10^{-6}$) following a cosine decay schedule. Setting $H = \lfloor (0.0075/\eta)^{-3} \rfloor$ results in consistently worse test accuracy than QSR (with the same communication volume) before epoch 265. However, during the final communication round, which spans from epoch 265 to 300, the cubic rule catches up with QSR. During this period, the learning rate dramatically decreases from $3.5 \times 10^{-4}$ to nearly zero, but our quasistatic view assumes that the learning rate $\eta_t$ should remain relatively constant for at least one communication round.

Based on the above observation, we argue that the cubic rule offers benefits over QSR only for certain schedules that have a rapid tail of learning rate decay near the end of training. To validate this view, we replace the cosine decay schedule with a variant of the step decay schedule in Smith et al. (2020). In our step decay schedule, given a total of 300 epochs, the learning rate remains at its peak until epoch 150, after which it is divided by 2 every 30 epochs. See Figure 7 for an illustration. Unlike the cosine schedule, this step decay schedule maintains a constant learning rate for a significant amount of time. As shown in Table 6(a), the cubic rule yields inferior generalization performance compared with our QSR, even after careful tuning of $\rho$. See Appendix C.2 for training details.

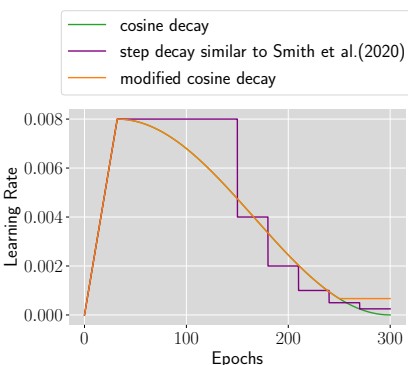

Figure 7: An illustration of the learning rate schedules.

Another way to corroborate our view is to run both scalings with a modified cosine learning rate schedule, which ceases to decay after a specific epoch $t''$ and remains constant until training ends. See Figure 7 for an illustration of this modified cosine schedule. As shown in Table 6(b), QSR consistently outperforms the cubic rule across various choices of $t''$. Further training details can be found in Appendix C.2. The probable reason is that when the learning rate is held constant, the cubic rule results in an excessively large $H$, negatively impacting optimization.

Given these failure cases of the cubic rule, we generally recommend using the QSR and leave it to future work to design a better rule to deal with schedules that have a rapid tail of learning rate decay.

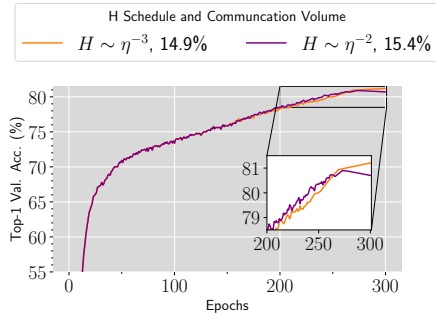

Figure 8: Test accuracy curves for QSR ($\alpha = 0.01$) and the cubic rule ($\rho = 0.0075$). The cubic rule results in consistently worse test accuracy than QSR (with the same communication volume) before the last communication round.

## H    COMPARISON WITH LOCAL SGD/ADAMW + SWAP

In this section, we compare QSR with the modified Stochastic Weight Averaging in Parallel (SWAP) algorithm, termed "Local SGD/AdamW + SWAP". Specifically, the original SWAP proposed by (Gupta et al., 2020) uses SGD for the majority of the training process and only switches to local updates at some $t_0$ near the end, thus saving less communication than QSR. To compare SWAP with QSR at a similar level of communication volume, we experiment with the modified SWAP, which starts with Local SGD/AdamW using a constant communication period $H_{\text{base}}$ and, after some time $t_0$, lets workers perform local updates with a final model averaging. As shown in Figure 9, QSR outperforms Local SGD/AdamW SWAP though we have tuned $t_0$ carefully for the latter.

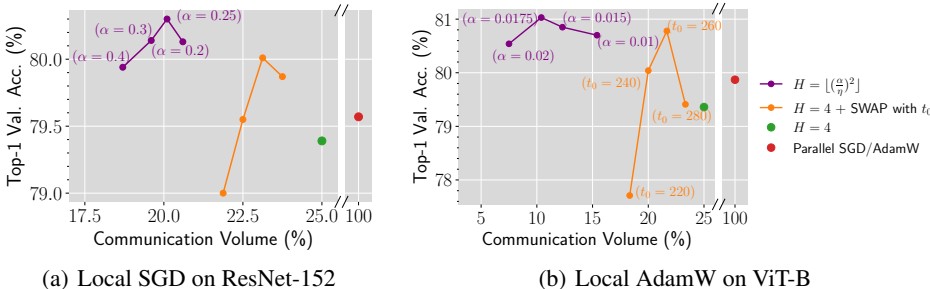

(a) Local SGD on ResNet-152

(b) Local AdamW on ViT-B

Figure 9: QSR outperforms Local SGD/AdamW + SWAP on both models. See Appendix C for training details.

