# OpenReview forum: "A Quadratic Synchronization Rule for Distributed Deep Learning"
_ICLR.cc/2024/Conference — ICLR 2024 poster_

### Official Review · Reviewer_duf3 · 2023-10-31

**Soundness:** 3 good
**Presentation:** 3 good
**Contribution:** 3 good
**Rating:** 8
**Confidence:** 2

**Summary:**

This work proposes a new synchronization rule for local gradient methods, where synchronization among processors is performed after multiple rounds of local updates. In the synchronization rule, the number of local updates is proportional to the inverse of the learning rate square. Theoretical analysis is provided to motivate the proposed synchronization rule. Experimental results on ResNet and ViT show that local gradient methods with the proposed synchronization rule yield better test accuracy and less training time.

**Strengths:**

1. The proposed quadratic synchronization rule is simple but effective, backed up by strong theoretical analysis.
2. Experimental results show both efficiency gain and better generalizations.
3. The experiments are comprehensive, including various architectures, learning rate decay strategies, and batch sizes.

Overall, I think this is a good paper.

**Weaknesses:**

n/a

**Questions:**

n/a

---

> ### Author Response · Authors · 2023-11-21
> **Response to Reviewer duf3**
>
> We sincerely thank the reviewer for the positive and insightful feedback on our paper. We are glad the reviewer found the proposed synchronization rule effective and supported by solid theoretical analysis and comprehensive experiments.
>
> We thank the reviewer again for the review and appreciation of our paper.

---

### Official Review · Reviewer_ydD5 · 2023-11-01

**Soundness:** 2 fair
**Presentation:** 2 fair
**Contribution:** 2 fair
**Rating:** 5
**Confidence:** 4

**Summary:**

The paper examines the optimal approach for dynamically determining H, a hyperparameter in Local SGD, allowing workers to compute locally for H steps without synchronizing with others.

The goal is to strike a balance between communication costs and the final test accuracy. While a large 'H' might degrade accuracy, a small 'H' could intensify communication overhead. Drawing inspiration from the stochastic differential equation introduced by [Gu et al.](https://arxiv.org/pdf/2303.01215.pdf), this paper introduces a new Quadratic Synchronization Rule for setting the H dynamically given a predetermined learning rate schedule. Furthermore, the research provides proof delineating the approximate convergence bounds concerning the distance between the iteration and the manifold. The paper also presents experimental results in various settings.

**Strengths:**

S1. The concept presented is intuitive—convergence can be improved by dynamically tuning H the number of steps of local updates before syncing with others. The introduced Quadratic Synchronization Rule holds potential for application in distributed machine learning.

S2. The paper provides a theoretical analysis of their idea and conducts experiments to support their theoretical results.

**Weaknesses:**

W1. The motivation of finding 'H’  given a learning rate schedule is questionable. Both learning rate and H should be tunable hyperparameters in distributed learning.
"Specifically, we introduce a simple yet effective strategy, called Quadratic Synchronization Rule (QSR), for dynamically adjusting the synchronization period according to the learning rate: given a learning rate schedule, we set H proportional to η^−2 as the learning rate η decays. This rule is largely inspired by a previous theoretical work (Gu et al., 2023), which shows that the generalization benefits arise only if H = Ω( 1/η ) when η → 0, but did not make any recommendation on how to set H."

Solely determining H given the learning rate schedule, yet without asking about what the optimal learning rate is is a less valuable question. While prior work [1] already shows that the generalization benefits arise only if H = Ω( 1/η ) when η → 0, this work specifically sets H to 1/η^2, which is incremental, reinforcing the same results. In fact, the learning rate is of greater significance in deep learning and distributed learning; for many problems, one can boost generalization and convergence performance solely by carefully tuning the learning rate. A more valid question that I believe should be asked is what is the optimal dynamic learning rate schedule under a given schedule of H, e.g. H=1 or 2 or Post-local SGD with variable H. There is no comparison to this learning rate question under a fixed H schedule, which further saves communication and is easier to implement in reality. Therefore, I think the problem studied here to recommend how to set H is kind of artificial.

W2. The baseline for comparison is not clear. For instance, what is the Parallel AdamW? How does the Parallel AdamW work? I can’t find the rigorous definition of it given by the paper. Because of the second momentum terms, Parallel AdamW that assumably aggregates the model parameters every H=1 step is not equivalent to Centralized AdamW when only setting H = 1. The second momentum can not be simply averaged. Even the definition of Parallel SGD is not provided. The paper simply says that refers to Local SGD when H=1. At least a reference or a definition should be provided.

W3. Some experimental settings are questionable. What has motivated the adoption of only SGD for ResNet experiments and AdamW only for VIT experiments? Furthermore, while all experiments employ momentum-based optimizers and underscore the effectiveness of QSR with such optimizers, the paper doesn't deeply explore the role of momentum in this methodology.

W4. The theoretical analysis and proof overlap too much with previous work [1]. Both rely on the same SDE technique (which is the novelty of the prior work) for analyzing analogous issues, with the current work lacking additional theoretical insights and development. Additionally, some proof steps are missing. See questions for details.

W5. There is no comparison with other practical distributed learning systems and methods, given the vast literature on distributed learning. The comparison is more of a simple self-comparison between different hyperparameter scalings of H, e.g., constant H vs. the QSR of H,  assuming the same learning rate schedule is used. This comparison is not convincing because 1) for different H setups, the optimal learning rate schedule will vary; 2) it didn’t compare to other possibilities of H = Ω( 1/η ) making it hard to understand the limitations of the problem domain. This makes the work more of hyperparameter tuning heuristics and lacks insights. The experimental results are more of validating the theorems and lack practical value.

W6. The exploration of the choice for $H_{base}$ seems insufficient. Providing a graph that illustrates the relationship between final test accuracy, communication volume, training time reduction, and the selection of $H_{base}$ would be more persuasive.

For the above reasons, I would recommend rejection.

Reference:

[1] Gu, Xinran, et al. "Why (and When) does Local SGD Generalize Better than SGD?." in ICLR 2023.

**Questions:**

(1) Have you considered comparing the performance of $H\sim \eta^{-3}$, $H\sim \eta^{-4}$? While prior work [Gu et al.](https://arxiv.org/pdf/2303.01215.pdf) already shows that the generalization benefits arise only if H = Ω( 1/η ) when η → 0, this work specifically sets H to $η^{-2}$. This results seem to be quite heuristic and not a thorough investigation of the problem domain. What understanding about the limitations or insights of the method has been derived?

(2) The proof provided for Lemma E.2 appears to be too succinct. Furthermore, in Lemma E.2, it is stated that 'we can apply the Itô-Taylor expansion from Lemma B.7 of Malladi et al. (2022) to derive (16), (17), and (18).' Is there a typo in this reference to equations? I suspect you meant to refer to equations (11), (12), and (13). Also, could you please elucidate how you derived equations 11, 12, and 13?

(3) Proof for Theorem 3.1 is overly concise. What is Lemma I.43? What is the definition of r? How do you get
$$E[u(\hat{\zeta}_{s, s+1},(s+1) \alpha^2, n \alpha^2)]$$

from

$$E[g(\hat{\zeta}_{s,n})]$$

(4) How do you calculate the communication time in Table 4, why don't you opt for using communication volume instead?

---

> ### Author Response · Authors · 2023-11-21
> **Response to Reviewer ydD5 (Clarification of Misunderstandings)**
>
> We sincerely thank the reviewer for the time and effort in reviewing our paper. However, in the first place, we would like to respectfully point out the misunderstandings the reviewer may have regarding our paper.
>
> **M1: The reviewer confuses optimization with generalization, leading to a misunderstanding of (1) how QSR works and (2) our theoretical results.**
>
> 1. The reviewer summarizes one of our strengths as “The concept presented is intuitive—convergence can be improved by dynamically tuning H the number of steps of local updates before syncing with others.”
>
> 2. The reviewer summarizes our theoretical results as “The research provides proof delineating the approximate convergence bounds concerning the distance between the iteration and the manifold.”
>
> **Clarification:** As highlighted in the introduction, due to the overparameterized nature of modern neural networks, reaching the same training loss does not correspond to having the same performance on test data. Our paper focuses on improving the test performance rather than optimizing the training loss.
>
> 1. QSR is designed for better generalization instead of faster convergence. Specifically, QSR dynamically sets $H$ in proportion to $\eta^{-2}$ to maintain the regularization effect of local steps as $\eta$ decays over time.
>
> 2. Our main theoretical contribution is the derivation of the slow SDE for Local SGD with QSR, which shows that QSR provably induces a stronger implicit bias towards flatter minima than other scalings, i.e., Local SGD with constant $H$ or $H\sim \eta^{-1}$, and Parallel SGD. By contrast, this paper does not provide “convergence bounds concerning the distance between the iteration and the manifold” mentioned by the reviewer.
>
> **M2: The reviewer mistakenly believed that we did not tune the learning rate**: “Solely determining H given the learning rate schedule, yet without asking about what the optimal learning rate is is a less valuable question. (in W1)” and “The comparison is more of a simple self-comparison between different hyperparameter scalings of H, e.g., constant H vs. the QSR of H, assuming the same learning rate schedule is used. (in W5)”
>
> **Clarification**: We have already experimented with the most widely used schedules [2-5], including cosine, linear, and step decay, and carefully optimized the learning rate. Specifically, for each baseline, i.e., Parallel SGD/AdamW and Local SGD/AdamW with a constant $H$, we have conducted a careful grid search on the peak learning rate, as stated in the “Training Setup” paragraph in Section 4 and Appendix C. For QSR, we adopt the same learning rate as their counterpart with a constant $H=H_{\mathrm{base}}$ in most cases, as detailed in Appendix C. Our experiments demonstrate QSR can yield consistent generalization benefits across different learning rate schedules.
>
> **M3: The reviewer mistakes the goal of our paper for “striking a balance between communication costs and the final test accuracy.”**
>
> **Clarification:** We aim to improve test accuracy and communication efficiency **at the same time**. This is justified by both our large-scale experiments and SDE-based theory.

---

> ### Author Response · Authors · 2023-11-21
> **Response to Reviewer ydD5 (Concerns & Questions, 1/3)**
>
> We believe many of the reviewer’s concerns are due to the above misunderstandings, and we hope the above clarification can help the reviewer understand our paper better. In the following, we address the reviewer’s other concerns and questions.
>
> **W1 (1)**: The motivation for finding 'H’ given a learning rate schedule is questionable. Solely determining H given the learning rate schedule, yet without asking about what the optimal learning rate is, is a less valuable question. What the optimal dynamic learning rate schedule is under a given schedule of H, e.g., H=1 or 2 or Post-local SGD with variable H, is a more valid question.
>
> **A**:
> 1. As clarified in our response to M2, we have explored the most popular and standard learning rate schedulers and carefully tuned the learning rate for each baseline algorithm, showing that QSR can yield consistent generalization benefits across different learning rate schedules.
>
> 2. Tuning learning rates for existing $H$ schedules (as suggested by the reviewer) cannot fully exploit the generalization benefit of local steps. According to our theory and experiments, to maintain the regularization effect, $H$ should grow to a very large value in the late training phase when the learning rate decays to a small value. For example, when training ViT-B with batch size 4096, $H$ can grow to over 8000 at the end of training. However, existing schedules of $H$ typically use very small values even in the late phase (e.g., the largest H the reviewer suggests seems to be 2).
>
> **W1 (2)**: While prior work [1] already shows that the generalization benefits arise only if $H=\Omega(\eta^{-1})$ when $\eta\to 0$, this work specifically sets $H$ to $\eta^{-2}$, which is incremental.
>
> **A**: QSR is novel compared to previous work [1]. Specifically, [1] focuses on the theoretical side, and only studies the setting of **using constant $\eta$ and $H$ throughout training**. No concrete $H$ schedules are proposed in [1]. By contrast, this paper proposes a specific $H$ schedule, QSR, where $H$ dynamically grows in proportion to $\eta^{-2}$ during training as $\eta$ decays over time. The setting of a decaying learning rate schedule studied in this paper is a common choice in practice (e.g., [2-4]), but it is not included in [1].
>
> **W2**: The baseline for comparison is not clear. What is Parallel AdamW/Parallel SGD?
>
> **A**: Parallel SGD/AdamW refers to the standard data parallel implementation of SGD/AdamW, which is introduced in the very first paragraph: at each step, each worker first computes gradients on their own local batches of data. Then, they average the local gradients via an All-Reduce operation and update the model parameter using the averaged gradient and SGD/AdamW optimizer. In the updated version, we have explicitly defined “Parallel OPT” as the data parallel implementation of an optimizer OPT in the first paragraph and included the pseudo-code for Parallel OPT in Appendix B according to the reviewer’s suggestion.
>
> **W3 (1)**:  Some experimental settings are questionable. What has motivated the adoption of only SGD for ResNet experiments and AdamW only for VIT experiments?
>
> **A**: Adopting SGD for ResNets and Adam/AdamW for ViTs is the standard practice (e.g., [5-10]). This enables us to run strong baselines with existing training recipes [6, 9] and showcase that QSR can be seamlessly integrated into standard training recipes for both convolutional and transformer-based models, enhancing both communication efficiency and generalization performance.
> It is also worth noting that no standard training recipe is known for training ViTs with SGD. Directly running SGD on transformer-based models typically results in a much slower convergence speed [15].
>
> **W3 (2)**: While all experiments employ momentum-based optimizers and underscore the effectiveness of QSR with such optimizers, the paper doesn't deeply explore the role of momentum in this methodology.
>
> **A**:
>
> 1. Adding momentum is an orthogonal direction to adjusting H for improving optimization and generalization. Our theoretical analysis builds upon the Slow SDE framework originally developed only for vanilla SGD [16]. This is novel and highly nontrivial. Understanding the role of momentum in this method is an interesting and challenging problem we plan to investigate in our future research.
>
> 2. Our experimental results on SGD/Local SGD with momentum align with the prediction of our theory, demonstrating that the implication of our theory goes beyond vanilla SGD/Local SGD.

---

> ### Author Response · Authors · 2023-11-21
> **Response to Reviewer ydD5 (Concerns & Questions, 2/3)**
>
> **W4**: The theoretical analysis and proof overlap too much with previous work [1].
>
> **A**: We respectfully disagree with the above comments. The analysis in [1] cannot handle the case where $H \sim \eta^{-2}]$, but our theoretical result can. Specifically, their analysis depends on the condition $H\eta=O(1)$, and many bounds therein explode as $H\eta \to \infty$, e.g., Theorem 3.3, Lemmas I.14 and I.16. In the context of QSR, where $H\eta=\frac{\alpha^2}{\eta}$ goes to infinity as $\eta$ approaches $0$, the condition $H\eta=O(1)$ is violated, rendering the analysis in Gu et al. (2023) ineffective for QSR. To help the reviewer better understand our proof, we have added a proof outline in the updated version.
>
> **W5**: There is no comparison with other practical distributed learning systems and methods. The comparison is more of a simple self-comparison between different scalings of $H$.  This comparison is unconvincing because 1) for different H setups, the optimal learning rate schedule will vary; 2) it didn’t compare to other possibilities of $H = \Omega(\frac{1}{\eta})$.
>
> **A**: We respectfully disagree with the reviewer’s interpretation of our empirical evaluation as a “self-comparison”.
>
> 1. One of our baselines, Parallel SGD/AdamW, is the common and standard algorithm for distributed training (e.g., [6, 7, 10, 11]). Another baseline, setting $H$ as a constant throughout training, is the conventional approach for local gradient methods (e.g., [12-14]). Therefore, we have compared QSR with standard and widely-used algorithms instead of making a self-comparison.
>
> 2. As clarified in our response to M2, we have explored the most popular and standard learning rate schedulers and carefully tuned the learning rate for each baseline algorithm, ensuring that our comparison is fair.
>
> 3. As for (2), we have already compared QSR with the scaling $H\sim \eta^{-1}$ in Figure 1 and Section 3. For completeness, we included $H\sim \eta^{-3}$ in Appendix G of the updated version, which shows that $H\sim \eta^{-3}$ does not provide consistent improvements over QSR.
>
> **W6**: The exploration of the choice for $H_{\mathrm{base}}$ seems insufficient.
>
> **A**:
>
> 1. Our paper has explored 2 choices of $H_\mathrm{base}$ for both ResNet and ViT across batch sizes of 4096 and 16384, which encompasses **8 settings**. As discussed in Section 4.2, the effect of $H_{\mathrm{base}}$ on the final test accuracy is consistent across all settings.
>
> 2. As elaborated in Sections 2 and 4.2, $H_{\mathrm{base}}$ should be determined based on the communication overhead. For our specific hardware, setting $H_{\mathrm{base}}$ as 2, 4 for Local SGD on ResNet-152 and as 4, 8 for ViT-B suffices to reduce the communication time to an inconsequential amount, taking up less than 12% of the total time (see Table 4).
>
> 3. Given the relatively large model size, exploring the 8 settings in our paper is already very expensive and time-consuming. For example, training ViT-B with batch size 4096 for 300 epochs takes 27 hours on 16 NVIDIA GeForce RTX 3090 GPUs.
>
> **Q1(1)**: Have you considered comparing the performance of QSR with $H\sim \eta^{-3}$ or $H\sim \eta^{-4}$? While prior work [1] already shows that the generalization benefits arise only if $H = \Omega( 1/\eta )$ when $\eta \to 0$, this work specifically sets $H$ to $\eta^{-2}$. This result seems to be quite heuristic and not a thorough investigation of the problem domain.
>
> **A**:
>
> 1. As clarified in our response to W1(2), QSR is novel compared to [1]. Specifically, [1] studies the setting of using **constant $\eta$ and $H$** throughout training. By contrast, in QSR, $H$ dynamically grows in proportion to $\eta^{-2}$ during training as $\eta$ decays over time. The setting of a decaying learning rate schedule studied in this paper is a common choice in practice (e.g., [2-4]) and not studied in [1].
> 2. QSR goes far beyond a heuristic, and its strong implicit bias for sharpness reduction is well supported by our theory. To explore other choices of power, our paper has already included a thorough comparison of QSR with $H\sim \eta^{-1}$ both theoretically and empirically. Following the reviewer’s suggestion, we included $H\sim \eta^{-3}$ in Appendix G of the updated version, which shows that $H\sim \eta^{-3}$ does not provide consistent improvements over QSR.
>
> **Q1(2)**: What understanding of the limitations or insights of the method has been derived?
>
> **A**: QSR offers an important insight that improved generalization performance and communication efficiency can be simultaneously achieved by adopting an appropriate $H$ schedule. The limitation of QSR, as discussed in Section 5, is that it may not consistently deliver noticeable generalization improvements for smaller models with shorter horizons. However, training in this regime is not costly, making it less of a critical concern.

---

> ### Author Response · Authors · 2023-11-21
> **Response to Reviewer ydD5 (Concerns & Questions, 3/3)**
>
> **Q2**: The proof provided for Lemma E.2 appears to be too succinct. Furthermore, in Lemma E.2, it is stated that 'we can apply the Itô-Taylor expansion from Lemma B.7 of Malladi et al. (2022) to derive (16), (17), and (18).' Is there a typo in this reference to equations? I suspect you meant to refer to equations (11), (12), and (13). Also, could you please elucidate how you derived equations (11), (12), and (13)?
>
> **A**: We thank the reviewer for pointing out the typo; we have corrected it in the updated version. As stated in the proof of Lemma E.2, to compute the moments of ${\phi}^{(s)}_{k, H}-{\phi}^{(s)}$ (eq (11)-(13) in the original submission and (13)-(16) in the updated version), it suffices to compute the moments of ${\zeta}(\alpha^2)-{\phi}^{(s)}$ since their moments differ in only $o(\mathrm{poly}(\alpha))$. The omitted step is a simple application of the standard Itô-Taylor expansion to the moments of ${\zeta}(\alpha^2)-{\phi}^{(s)}$. To provide the reviewer with a clearer understanding of our proof, we have enhanced the proof of Lemma E.2 with additional details and included the itô-Taylor expansion Lemma in the updated version.
>
> **Q3**: Proof for Theorem 3.1 is overly concise. What is Lemma I.43? What is the definition of $r$? How do you get
>
> $\mathbb{E}[u(\hat{{\zeta}}\_{s, s+1}, (s+1)\alpha^2, n\alpha^2)]$ from  $\mathbb{E}[g(\hat{\zeta}\_{s, n})]$?
>
> **A**: Following the reviewer’s suggestion, we have enhanced the proof of Theorem 3.1 with additional details. Below, we answer the reviewer’s questions one by one.
>
> 1. $r$ is some integer greater than or equal to $s$.
>
> 2. We can connect $\mathbb{E}[u(\hat{{\zeta}}\_{s, s+1}, (s+1)\alpha^2, n\alpha^2)]$ with $\mathbb{E}[g(\hat{{\zeta}}\_{s, n})]$ by a simple application of the law of total expectation. Please see the updated version for details.
>
> 3. Now, we elucidate the omitted steps that are similar to Lemma I.43 of [1]. Notice that to bound $\mathbb{E}[|g(\phi^{(n)})] - \mathbb{E}[g({\zeta}(n\alpha^2))]|$, it suffices to bound $\mathbb{E}[u({\phi}^{(s+1)}, (s+1)\alpha, n\alpha^2)] - \mathbb{E}[u(\hat{{\zeta}}\_{s, s+1}, (s+1)\alpha, n\alpha^2  )]$, where the two terms only differ in the first position. As stated in the proof, we already know that the moments of ${\phi}^{(s+1)}-{\phi}^{(s)}$ and $\hat{{\zeta}}\_{s, s+1}-{\phi}^{(s)}$ are close to each other. The problem reduces to discussing the smoothness of $u$ and performing Taylor expansion.
>
> **Q4**: How do you calculate the communication time in Table 4? Why not opt for using communication volume instead?
>
> **A**: We derive the communication time from the difference in total training time across runs with various communication frequencies since the asynchronous nature of CUDA makes it hard to directly measure the communication time. Please refer to Appendix F in the updated version for details. Most of our figures and tables have already reported the communication volume, e.g., Figures 1 to 3, and Tables 1 to 3. The wall-clock time analysis aims to showcase that, compared with Parallel SGD/AdamW, QSR can indeed accelerate training by significantly reducing communication overhead.
>
> We thank the reviewer again for the time and effort in reviewing our paper. We hope that we have addressed the key concerns of the reviewer. If so, we kindly ask the reviewer to raise the rating. If the reviewer has any further questions, we will be more than happy to answer them.

---

> ### Author Response · Authors · 2023-11-21
> **Response to Reviewer ydD5 (References)**
>
> [1] Xinran Gu, Kaifeng Lyu, Longbo Huang, and Sanjeev Arora. Why (and when) does local SGD generalize better than SGD? In The Eleventh International Conference on Learning Representations, 2023.
>
> [2] Ze Liu, Yutong Lin, Yue Cao, Han Hu, Yixuan Wei, Zheng Zhang, Stephen Lin, and Baining Guo. Swin transformer: Hierarchical vision transformer using shifted windows. In Proceedings of the IEEE/CVF international conference on computer vision, pp. 10012–10022, 2021.
>
> [3] Zhuang Liu, Hanzi Mao, Chao-Yuan Wu, Christoph Feichtenhofer, Trevor Darrell, and Saining Xie. A convnet for the 2020s. In Proceedings of the IEEE/CVF conference on computer vision and pattern recognition, pp. 11976–11986, 2022.
>
> [4] Peter Izsak, Moshe Berchansky, and Omer Levy. How to train bert with an academic budget. arXiv preprint arXiv:2104.07705, 2021.
>
> [5] Kaiming He, Xiangyu Zhang, Shaoqing Ren, and Jian Sun. Deep residual learning for image recognition. In Proceedings of the IEEE conference on computer vision and pattern recognition, pp. 770–778, 2016.
>
> [6] Lucas Beyer, Xiaohua Zhai, and Alexander Kolesnikov. Better plain vit baselines for imagenet-1k. arXiv preprint arXiv:2205.01580, 2022.
>
> [7] Alexey Dosovitskiy, Lucas Beyer, Alexander Kolesnikov, Dirk Weissenborn, Xiaohua Zhai, Thomas Unterthiner, Mostafa Dehghani, Matthias Minderer, Georg Heigold, Sylvain Gelly, Jakob Uszkoreit, and Neil Houlsby. An image is worth 16x16 words: Transformers for image recognition at scale. In International Conference on Learning Representations, 2021.
>
> [8] Hugo Touvron, Matthieu Cord, Matthijs Douze, Francisco Massa, Alexandre Sablayrolles, and Hervé Jégou. Training data-efficient image transformers & distillation through attention, 2021b.
>
> [9] Pierre Foret, Ariel Kleiner, Hossein Mobahi, and Behnam Neyshabur. Sharpness-aware minimization for efficiently improving generalization. In International Conference on Learning Representations, 2021a.
>
> [10] Priya Goyal, Piotr Dollár, Ross Girshick, Pieter Noordhuis, Lukasz Wesolowski, Aapo Kyrola, Andrew Tulloch, Yangqing Jia, and Kaiming He. Accurate, large minibatch SGD: Training imagenet in 1 hour. arXiv preprint arXiv:1706.02677, 2017.
>
> [11] Ze Liu, Yutong Lin, Yue Cao, Han Hu, Yixuan Wei, Zheng Zhang, Stephen Lin, and Baining Guo. Swin transformer: Hierarchical vision transformer using shifted windows. In Proceedings of the IEEE/CVF international conference on computer vision, pp. 10012–10022, 2021.
>
> [12] Sebastian U Stich. Local SGD converges fast and communicates little. In International Conference on Learning Representations, 2018.
>
> [13] Hao Yu, Sen Yang, and Shenghuo Zhu. Parallel restarted SGD with faster convergence and less communication: Demystifying why model averaging works for deep learning. In Proceedings of the AAAI Conference on Artificial Intelligence, volume 33, pp. 5693–5700, 2019.
>
> [14] Jianyu Wang, Vinayak Tantia, Nicolas Ballas, and Michael Rabbat. Slowmo: Improving communication-efficient distributed SGD with slow momentum. In International Conference on Learning Representations, 2019.
>
> [15] Jingzhao Zhang, Sai Praneeth Karimireddy, Andreas Veit, Seungyeon Kim, Sashank Reddi, Sanjiv Kumar, and Suvrit Sra. Why are adaptive methods good for attention models? Advances in Neural Information Processing Systems, 33:15383–15393, 2020.
>
> [16] Zhiyuan Li, Tianhao Wang, and Sanjeev Arora. What happens after SGD reaches zero loss?–a mathematical framework. In International Conference on Learning Representations, 2021c.

---

### Official Review · Reviewer_tFqJ · 2023-11-01

**Soundness:** 3 good
**Presentation:** 2 fair
**Contribution:** 3 good
**Rating:** 6
**Confidence:** 3

**Summary:**

This paper builds upon the theoretical insights from [[1]](https://openreview.net/pdf?id=svCcui6Drl) to introduce the "Quadratic Synchronization Rule" (QSR), a theoretically grounded method to set the parameter $H$ in Local-SGD. By analyzing a "Slow SDE for Local SGD with QSR", it is shown that scaling $H \propto \frac 1 {\eta^2}$ (with $\eta$ the Local-SGD decaying learning rate) allows to regularize the objective function towards flatter minima, increasing generalization compared to other schemes (constant values of $H$, or $H \propto \frac 1 {\eta}$). Extensive experiments using ResNet-152 and ViT-B on ImageNet are made to confirm the advantages of QSR compared to previous methods both in terms of test accuracy and wall-clock time for the training.

[1] Xinran Gu and Kaifeng Lyu and Longbo Huang and Sanjeev Arora. *Why (and When) does Local {SGD} Generalize Better than {SGD}?*, In The Eleventh International Conference on Learning Representations, 2023.

**Strengths:**

* **Promising experimental results**: In the presented settings, the introduction of QSR seems to lead to noticable gains in terms of validation accuracy compared to all the chosen baselines, as well as some mild savings in wall-clock time compared to methods using a constant $H$.
* **Theoretical analysis of Slow SDE for Local SGD with QSR**: Efforts are made to make the reader intuitively understand the changes the introduction of QSR makes to the Slow SDE and how the $K$ times larger drift term favors generalization.

**Weaknesses:**

* **No confidence intervals in experiments**: given the extensive hyper-parameter search Appendix C claims, I would have expected that the  experimental results in the main paper would come with confidence intervals, and were presented as average over several runs to confirm the reproducibility of the results.
* **No visualization of the effective $H$ scheduler**: Tab. 4 hints that using constant $H$ and QSR actually leads to very similar training times, suggesting that using the QSR scheduler for the values of $\alpha$ retained does not lead to major deviations compared to using constant $H$. In effect, what does the evolution of $H$ look like throughout training, and how different is it from using a constant $H$ ?
* **Theoretical reasons for the focus on large model and long horizon unclear**: While the main paper focuses on fairly large models (ResNet-152 and ViT-B) and long horizon (200-300 epochs on ImageNet), Appendix D shows that for training a ResNet-50 on 90 epochs, QSR has no effect. Are there any theoretical reasons why QSR should only work for large model and training procedures ?
* **Unclear why $\gamma =2$ is optimal:** Section 2 claims that *"$H^{(s)}$ could have been set to $H^{(s)}:= \max \left \\{ H_\text{base},  \left \lfloor \left ( \frac{\alpha}{\eta_t}  \right )^{\gamma} \right \rfloor \right \\}$ for any $\gamma$"*. However, it still remains unclear to me why setting $\gamma = 2$ is optimal.
* **Comparison with related work lacking:** Appendix A cites the *"Stochastic Weight Averaging in Parallel (SWAP) algorithm"* [[2]](https://openreview.net/pdf?id=rygFWAEFwS) as another method that varies the value of $H$ during training, leading to a decrease in training time and better generalization. Yet, no comparison with the method is performed in the paper.
* **Additional hyper-parameter to tune**: *"choosing the right value of $H$"* has been traded for *"choosing the right value of $\alpha$"*, thus QSR does not "automatically" set $H$ and no hyper-parameter tuning has been saved.

[2] Vipul Gupta and Santiago Akle Serrano and Dennis DeCoste. *Stochastic Weight Averaging in Parallel: Large-Batch Training That Generalizes Well*, In International Conference on Learning Representations, 2020.

**Questions:**

* A network speed of 25Gbps is used for the experiments. Would your algorithm still help in HPC settings where fast communications can happen over 100 Gbps Infiniband ?
* Are both the model size **and** number of epochs important factors for QSR to work ? For example, using a smaller model (such as ResNet18) but with a decent amount of epochs (e.g., 300 epochs on CIFAR10), would QSR have an impact ?

---

> ### Author Response · Authors · 2023-11-21
> **Response to Reviewer tFqJ (1/2)**
>
> We sincerely thank the reviewer for acknowledging our experimental results as promising and our theory as intuitive. Below we address the reviewer’s concerns and questions one by one.
>
> **Concern 1**: No confidence intervals in experiments.
>
> **A**: We have repeated our main experiments and updated Table 1. We are unable to repeat all experiments due to high training costs (~20 hours for a single run on 16 NVIDIA GeForce RTX 3090 GPUs), but the improvement is consistent across different settings (Tables 2 and 3, Figures 2 and 3).
>
> **Concern 2**: No visualization of the $H$ scheduler. How different is the $H$ schedule for QSR from using a constant $H$?
>
> **A**: Following the reviewer's suggestion, we have added a visualization (Figure 5) in the appendix, showing that it differs greatly from using a constant $H$. For example, in Figure 5, when we use $\alpha=0.0175$ for a 300-epoch cosine learning rate schedule with peak learning rate 0.008, $H$ starts to grow to 30 at epoch 183, and reaches 8142 at epoch 273 when$\eta$ decays to $1.9\times 10^{-4}$. Indeed, the difference in the $H$ schedule is significant enough to improve the test accuracy from 79.32% to 80.98%.
>
> **Concern 3**: Why focus on large models and long horizons? Are there any theoretical reasons for the effectiveness of QSR in this setup? (Weakness 2 and Question 2)
>
> **A**:
>
> 1. Large models are typically coupled with long horizons (e.g., several hundred epochs on ImageNet [3, 4, 8, 9]) to fully exploit the model’s capability. Training large models with long horizons is the setting where distributed training is necessary, and the computation cost is notoriously high, with the communication cost further exacerbating the situation. In our experiments, we follow standard training recipes of relatively large models, ResNet-152 [3] and ViT-B [4], and showcase that QSR can reduce communication costs and improve test accuracy.
>
> 2. By contrast, the small-scale experiments, e.g., ResNet-18 on CIFAR-10 mentioned by the reviewer, are cheap and fast, and do not require distributed training, let alone the communication cost.
>
> 3. The theoretical reasons are twofold. First, QSR is inspired by the Slow SDE, which, as interpreted in Section 3, captures the long-term dynamics of SGD/Local SGD. Ablation studies in Section 2 of [1] also show that the training budget should include enough iterations for the generalization benefits of Local SGD to arise, and that the generalization improvement over SGD becomes more significant as we train longer. Second, the model should be overparamterized, where the local minimizers of the training loss can form a manifold [2, 5, 6].
>
> **Concern 4**: Unclear why $\gamma=2$ is optimal in $H^{(s)}:=\max\bigg(\{H_{\mathrm{base}}, \lfloor (\frac{\alpha}{\eta_t})^{\gamma}\rfloor}\bigg)$.
>
> **A**: As shown in eq (3) of our paper, the drift term gets stronger as $H$ increases, driving the iterate to drift faster towards flatter minima. However, $H$ cannot grow arbitrarily large; otherwise, the slow SDE loses track of Local SGD, as reflected in the error bound in Theorem 3.1 (3) and Theorem 3.3 in [1]. Therefore, to optimize the generalization performance, we should set $H$ to the largest value without making the slow SDE approximation ineffective. $H\sim \eta^{-2}$ is the largest possible $H$ that one can find a valid slow SDE approximation near the minimizer manifold. Furthermore, we empirically demonstrate the optimality of $\gamma=2$ via comparisons with $\gamma=1$ (Section 3) and $\gamma=3$ (Appendix G in the updated version).
>
> **Concern 5**:  Missing comparison with SWAP [7].
>
> **A**: We have added the comparison with SWAP in Appendix H of the updated version, where we showcase QSR's superior generalization performance over SWAP. Specifically, the original SWAP proposed in [7] is less communication efficient than QSR since it uses SGD for the majority of the training process and only switches to local updates at some $t_0$ near the end. To compare SWAP with QSR at a similar level of communication volume, we experiment with a slightly modified SWAP, which starts with Local SGD/AdamW using a constant communication period $H_{\mathrm{base}}$ and, after some time $t_0$, lets workers perform local updates with a final model averaging. As shown in Figure 9, QSR outperforms SWAP, though we have tuned $t_0$ carefully for the latter.
>
> **Concern 6**: Additional hyperparameter to tune: “choosing the right value of $H$” has been traded for "choosing the right value of “$\alpha$,” thus QSR does not "automatically" set $H$ and no hyper-parameter tuning has been saved.
>
> **A**: We did not claim that QSR can ease the burden of hyperparameter tuning. Rather, we would like to highlight that, compared with $H$-schedules in previous works, e.g., post-local SGD, QSR improves test accuracy without increasing the number of hyperparameters to tune.

---

> ### Author Response · Authors · 2023-11-21
> **Response to Reviewer tFqJ (2/2)**
>
> **Question**: A network speed of 25Gbps is used for the experiments. Would your algorithm still help in HPC settings where fast communications can happen over 100 Gbps Infiniband ?
>
> **A**: The communication time is expected to be 1/4 of that reported in Table 4 as the bandwidth is increased to 4 times.  However, we do not know the exact speedup for our experiments over 100 Gbps Infiniband since we do not have access to such a network.
>
> We thank the reviewer again for the helpful feedback. We hope that we have addressed the key concerns of the reviewer. If so, we kindly ask the reviewer to raise the rating. If the reviewer has any further questions, we will be more than happy to answer them.
>
> **References**
>
> [1] Xinran Gu, Kaifeng Lyu, Longbo Huang, and Sanjeev Arora. Why (and when) does local SGD generalize better than SGD? In The Eleventh International Conference on Learning Representations, 2023.
>
> [2] Zhiyuan Li, Tianhao Wang, and Sanjeev Arora. What happens after SGD reaches zero loss?–a mathematical framework. In International Conference on Learning Representations, 2021c.
>
> [3] Pierre Foret, Ariel Kleiner, Hossein Mobahi, and Behnam Neyshabur. Sharpness-aware minimization for efficiently improving generalization. In International Conference on Learning Representations, 2021a.
>
> [4] Lucas Beyer, Xiaohua Zhai, and Alexander Kolesnikov. Better plain vit baselines for imagenet-1k. arXiv preprint arXiv:2205.01580, 2022.
>
> [5] Yaim Cooper. The loss landscape of overparameterized neural networks. arXiv preprint arXiv:1804.10200, 2018.
>
> [6] Felix Draxler, Kambis Veschgini, Manfred Salmhofer, and Fred Hamprecht. Essentially no barriers in neural network energy landscape. In International conference on machine learning, pp. 1309–1318. PMLR, 2018.
>
> [7] Vipul Gupta, Santiago Akle Serrano, and Dennis DeCoste. Stochastic weight averaging in parallel: Large-batch training that generalizes well. In International Conference on Learning Representations, 2020.
>
> [8] Ze Liu, Yutong Lin, Yue Cao, Han Hu, Yixuan Wei, Zheng Zhang, Stephen Lin, and Baining Guo. Swin transformer: Hierarchical vision transformer using shifted windows. In Proceedings of the IEEE/CVF international conference on computer vision, pp. 10012–10022, 2021.
>
> [9] Zhuang Liu, Hanzi Mao, Chao-Yuan Wu, Christoph Feichtenhofer, Trevor Darrell, and Saining Xie. A convnet for the 2020s. In Proceedings of the IEEE/CVF conference on computer vision and pattern recognition, pp. 11976–11986, 2022.

---

> > ### Comment · Reviewer_tFqJ · 2023-11-23
> >
> > I thank the authors for their detailed response.
> >
> > In light of the explanation for the reason behind the choice $H \propto \frac 1 {\eta^2}$, the  added experiments, as well as the new evidence that, in practice, the introduced scheduler for $H$ greatly differs from established strategies (Figure 5), I raise my score.

---

> > > ### Author Response · Authors · 2023-11-23
> > > **Thank you**
> > >
> > > Thank you for your support and timely response!

---

### Official Review · Reviewer_u85z · 2023-11-01

**Soundness:** 3 good
**Presentation:** 3 good
**Contribution:** 3 good
**Rating:** 8
**Confidence:** 3

**Summary:**

This paper proposes a scheduler for the synchronization interval of local SGD/ADAM, a.k.a. the Quadratic Synchronization Rule (QSR), which recommends dynamically setting such intervals in proportion to the inverse of the square of learning rate. The proposed algorithm is supported by theoretical analysis based on SDE. The empirical results show that QSR can achieve better validation accuracy with less communication overhead.

**Strengths:**

1. This paper proposes a scheduler for the synchronization interval of local SGD/ADAM, a.k.a. the Quadratic Synchronization Rule (QSR), which recommends dynamically setting such intervals in proportion to the inverse of the square of learning rate.

2. The proposed algorithm is supported by theoretical analysis based on SDE.

3. The empirical results show that QSR can achieve better validation accuracy with less communication overhead.

**Weaknesses:**

1. The experiments focuses on vision tasks and models. Although ViT model uses the transformer-like architecture, I'm still very interested in how QSR performs in NLP tasks with transformer architectures.

2. The theoretical analysis only applies to local SGD. I wonder whether it could be extended to local Adam.

**Questions:**

1. Although QSR is supported by theoretical analysis, I still wonder how the other options work, such as setting $H$ proportion to $\frac{1}{\eta}$ or $\frac{1}{\sqrt{\eta}}$. Is there any corresponding ablation experiments?

2. Many theoretical analysis suggests a learning rate scheduler $\eta \propto \frac{1}{\sqrt{T}}$, where $T$ is the number of steps. What if we detach the choice of $H$ from the learning rate, and simply use $H \propto T$ (with some rounding of course)?

3. Is it possible to extend the theoretical analysis to local Adam?

---

> ### Author Response · Authors · 2023-11-21
> **Response to Reviewer u85z**
>
> We sincerely thank the reviewer for the positive and constructive feedback. Below we address the reviewer’s questions.
>
> **Q1**: How does QSR perform in NLP tasks with transformer architectures?
>
> **A**: Thank you for your suggestion on exploring QSR on NLP tasks. As discussed in Section 5, local gradient methods may have the potential to improve the transferability from pretraining to downstream tasks in NLP. However, given that the large-scale experiments on vision tasks are already costly and time-consuming (~20 hours for a single run), we leave the exploration of QSR on NLP tasks for future work.
>
> **Q2**: Is it possible to extend the theoretical analysis to Local Adam?
>
> **A**: It is possible but very challenging. Though QSR is motivated by the slow SDE for SGD/Local SGD, it can also improve the test accuracy when applied to Local AdamW, indicating that there may be some similar implicit bias near the minimizer manifold for Local AdamW/Adam. However, notice that the analyses for Local SGD in our paper and [1] has already been quite complicated. Extending this to Local Adam/AdamW requires the characterization of the dynamics of the precondition matrix and momentum, further complicating the proof. Given these difficulties, we leave the analysis of Local Adam/AdamW to future work.
>
> **Q3**: How does QSR compare to other scalings such as $H\sim \eta^{-1}$ and $H\sim 1/\sqrt{\eta}$ ? Are there any ablation studies?
>
> **A**: Our paper already includes the experimental comparison of QSR and $H\sim\eta{-1}$ in Figure 2. To further validate the optimality of QSR, we conduct additional experiments on $H\sim \eta^{-3}$ in Appendix G of the updated version.
>
> **Q4**: Many theoretical analysis suggests a learning rate scheduler $\eta \propto \sqrt{\frac{1}{T}}$, where $T$ is the number of steps. What if we detach the choice of $H$ from the learning rate and simply use $H\propto T$?
>
> **A**: You are correct in pointing out that $H$ is proportional to $T$ if QSR is applied to a square root learning rate decay scheduler. However, this relationship does not universally apply to other widely used schedulers, e.g., cosine, linear and step decay, where $\eta\propto \sqrt{\frac{1}{T}}$ does not hold. Hence, we choose to express $H$ as the function of $\eta$, which offers a general form remaining consistent across different learning rate schedulers.
>
> We thank the reviewer for the appreciation of our paper and the constructive feedback!
>
> **References**
>
> [1] Xinran Gu, Kaifeng Lyu, Longbo Huang, and Sanjeev Arora. Why (and when) does local SGD generalize better than SGD? In The Eleventh International Conference on Learning Representations, 2023.

---

### Author Response · Authors · 2023-11-22
**Note on major revisions**

We sincerely thank all reviewers for their helpful feedback! Following the reviewers’ suggestions, we have made the revisions listed below:
- In Table 1, we reported the validation accuracy and train loss averaged over 3 runs, along with the standard deviation.
- We added Figure 5 to visualize the $H$ schedule for QSR.
- In Appendix B, we presented the pseudocode for standard data parallel methods.
- In Appendix E, we added a proof outline and two preliminary Lemmas. We also enhanced the proof for Lemma E.2 and Theorem 3.1 with more details.
- In Appendix F, we presented the details for communication time measurement.
- In Appendix G, we explored the scaling $H\sim \eta^{-3}$.
- In Appendix H, we presented experiments on a slightly modified Stochastic Weight Averaging in Parallel (SWAP) algorithm.

---

### Meta-Review · Area_Chair_fHWH · 2023-12-06

**Metareview:**

The paper proposes a new rule for choose how often local gradient descent should communicate. The rule is supported both by simulations, showing improved generalization with fewer communications, as well as by some theory. The theory here seems to be based on setting parameters to make a certain "slow SDE" approximation work. The results are interesting and the general consensus of the reviewers favors publication.

**Justification For Why Not Higher Score:**

As I understand it, the result proposed here is not fully rigorous in the sense that it is not proved that the rule proposed here is better than a number of possible alternative rules.

**Justification For Why Not Lower Score:**

The paper makes a clear and valuable contribution to the distributed optimization literature.

---

### Decision · Program_Chairs · 2024-01-16

Accept (poster)